# The social costs of tropical cyclones

Hazem Krichene[1], Thomas Vogt ⓘ[1], Franziska Piontek[1], Tobias Geiger[1,2], Christof Schötz[1] & Christian Otto ⓘ[1]✉

Tropical cyclones (TCs) can adversely affect economic development for more than a decade. Yet, these long-term effects are not accounted for in current estimates of the social cost of carbon (SCC), a key metric informing climate policy on the societal costs of greenhouse gas emissions. We here derive temperature-dependent damage functions for 41 TC-affected countries to quantify the country-level SCC induced by the persistent growth effects of damaging TCs. We find that accounting for TC impacts substantially increases the global SCC by more than 20%; median global SCC increases from US$ 173 to US$ 212 per tonne of $CO_2$ under a middle-of-the-road future emission and socioeconomic development scenario. This increase is mainly driven by the strongly TC-affected major greenhouse gas emitting countries India, USA, China, Taiwan, and Japan. This suggests that the benefits of climate policies could currently be substantially underestimated. Adequately accounting for the damages of extreme weather events in policy evaluation may therefore help to prevent a critical lack of climate action.

Tropical cyclones (TCs) are among the most harmful extreme weather events. They affect on average 20.4 million people annually, and they caused mean direct annual economic losses of US$ 51.5 billion averaged over the last decade[1]. Critically, there is increasing empirical evidence that TC impacts can reduce economic growth in the affected country for more than a decade[2–4]. In consequence, the economic repercussions of subsequent events can overlap in TC-prone countries, leaving insufficient time for the economy to recover in between[5,6]. In this way, the long-term reductions in economic growth may harm economic development even more strongly than the direct loss and damage caused by the TC landfalls[7–9]. The projected increase in the frequency of the most intense TCs under global warming[10,11] may render overlapping economic repercussions more likely, amplifying long-term growth losses in the absence of additional adaptation measures. This may deteriorate the development perspectives, especially for strongly affected low- and middle-income countries[3,12,13]. (Throughout the study, we group countries according to the 2024 World Bank's country income classification scale into high-income (2022 per–capita gross national income (GNIpc) US$≥13, 846), upper-middle-income (US$4, 466≤GNIpc≤US$13, 845)), lower-middle-income (US$1, 136≤GNIpc≤US$4, 465), and low-income countries (GNIpc≤US$1, 135)).

Over the years, a number of TC damage functions were developed, ranging from rather simple estimations of how damages change with global mean temperature (GMT) and socioeconomic development[14,15] to more complex event-based approaches accounting for characteristics of individual storms such as areas affected by strong TC winds[16] and lifetime rainfall[17]. Many works are USA specific[18–23] or employ damage functions derived for the USA to estimate damages in other countries[24,25]. Considerably fewer studies derive damage functions accounting for the TC climatology and socioeconomic vulnerabilities of other TC-prone countries such as the Philippines[26,27], China[28], or South-Korea[17], or to derive global sets of region-specific damage functions[29,30]. With the notable exceptions of Hsiang et al.[4] and Elliott et al.[25], projections of future damages in the literature are based on damage functions which statistically link TC predictors to reported (direct) damages. This approach has the advantage that damage databases can be employed for calibration, but it does not allow to account for the persistence of damages in the economic system. This could result in an underestimation of (future) damages and limit the applicability of these earlier estimates for national policy-makers and actors of international climate finance charged with estimating adaptation needs at the country level and comparing them across countries.

[1]Potsdam Institute for Climate Impact Research, Potsdam, Germany. [2]Deutscher Wetterdienst (DWD), Climate and Environment Consultancy, Potsdam, Germany. ✉e-mail: christian.otto@pik-potsdam.de

We here present a modeling framework to quantify future TC damages that comprises three components. First, we estimate the historical growth responses of countries to TC strikes accounting for the persistence of damages in the economic system. Second, we generate probabilistic, event-based projections of TC damages by projecting the historical growth losses along a large set of synthetic time series of TCs with landfalls generated with a TC emulator[31]. Accounting for different emission (Representative Concentration Pathways (RCPs)[32]), socioeconomic development (Shared Socioeconomic Pathways (SSPs)[33,34]), and economic discounting scenario combinations thereby allows for a systematic assessment of compounding uncertainties. Third, we derive event-based, and country-specific temperature-dependent damage functions that account for the persistence of damages in the economic system from the growth losses calculated along the different scenario combinations. Similarly to earlier works by Mendelsohn et al. [35] and Bakkensen et al.[17], we employ large sets of synthetic TCs with landfall to account for changes in TC climatology with global warming. In addition, we here account for the persistence of damages in the economic system, building on earlier works by Elliott et al.[25] and Hsiang et al.[4]. We finally express damages as functions of GMT change. These temperature damage functions are needed by most climate integrated assessment models to weigh the costs of mitigation and adaptation measures with the remaining climate change impacts[15]. In our analyses, we focus on two complementary metrics of high policy relevance: i) the discounted annual damage (DAD) caused by TCs and ii) the contribution of TCs to the social cost of carbon (SCC). Both metrics are projected for different RCP-SSP scenario combinations and discounting choices in order to allow for a thorough sensitivity assessment. The DAD is a measure for the additional TC-induced future economic burdens of countries induced by climate change. It reveals adaptation needs and allows the development of tailored evidence-based National Adaptation Plans (NAPs)[36]. We calculate DAD for 41 TC-prone countries and – given our

globally harmonized approach – compare the DAD among these countries. The SCC allows to directly measure the costs arising for the global society from additional $CO_2$ emissions which is why it is a key metric when it comes to informing the international climate negotiations and national mitigation decisions[37]. However, common approaches to estimate the SCC have been criticized for not adequately accounting for the contributions of extreme weather events[38,39]. Our SCC estimates are based on Ricke et al.[40]. This approach allows us to calculate the SCC globally as well as the contributions of individual countries. While the original estimates by Ricke et al. do not comprise the contributions of extreme weather events, we here employ our event-based temperature-dependent damage functions to quantify the contribution from TCs to the overall SCC.

## Results

### Historical growth responses of countries to tropical cyclones

We empirically estimate the long-term response of economic growth rates to TC strikes for a sample of 41 TC-affected countries over the historical period 1981–2015 (Fig. 1a). To avoid the potential endogeneity issues of earlier works using Barro-type growth regressions (see ref. 3 for a detailed discussion), we employ a three-way fixed effects panel model with annual national shares of people exposed to TCs for different lag-years as exogenous predictors introduced in ref. 2. We add methodologically to this previous work in two regards. First, we additionally control for the impact of temperature on growth. Thereby, we follow the approach of Burke et al.[41] and account for linear and quadratic terms in population-weighted annual mean temperatures over land for a set of 174 countries that includes the 41 TC-affected countries. Estimating temperature and TC impacts on growth in a single regression framework allows also to investigate potential interactions of both effects. If temperature and TCs can be considered as independent impact channels, their contributions to growth losses become additive. We find that including TC effects does not affect the

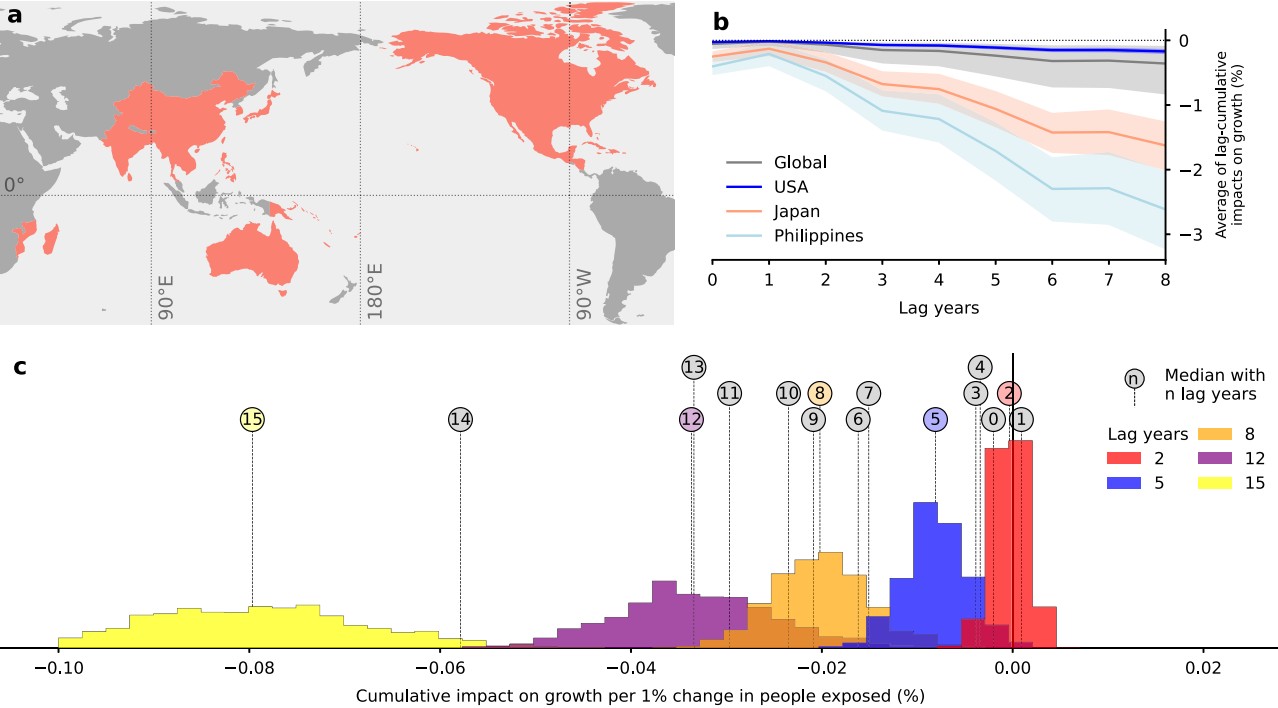

**Fig. 1 | Long-term economic growth response to tropical cyclones. a** Map of the set of 41 countries that are considered to be exposed to tropical cyclones in this analysis. **b** Average historical responses of per-capita GDP growth to tropical cyclone exposure as a function of lag years for selected countries (colors) and on global average (gray). Shaded areas indicate the 66% confidence intervals (as obtained from 1200 bootstraps (uncertainty dimension 1 in Table 1; $n = 42,000$ data points per country (Methods)). **c** Distributions of growth responses for model configurations with 2, 5, 8, 12, and 15 lag years (color code). Vertical lines with circles denote the medians of the distributions with 0–15 lag years ($n = 42,000$ data points per country).

**Table 1 | The six compounding uncertainty dimensions (UD) covered in the projections of tropical cyclone (TC) damages**

| Method | Uncertainty dimension (UD) | Coverage |
|---|---|---|
| Scenario analysis | Future greenhouse gas emissions (**UD1**) | A "no further climate change" baseline scenario and 3 representative concentration pathways (RCPs)[32]: RCP2.6, a Paris-compatible strong mitigation scenario where global GHG emissions already peak in 2020, RCP6.0, a "business-as-usual" scenario with peak emissions in 2080, and RCP8.5, a strong emission scenario where emissions continuously increase until 2100. |
| | Socioeconomic development (**UD2**) | 2 Shared Socioeconomic Pathways (SSPs)[34]: SSP2, a middle-of-the-road scenario with regard to economic and population growth, and SSP5, a strong fossil-fuel-based economic growth scenario with comparably low population growth. |
| | Normative assumptions on discounting (**UD3**) | 3 scenarios on how society weighs future compared to present TC damages. Choices of growth adjusted discount rate $\rho + \eta g$, where $\rho$, $\eta$ and $g$ denote the rate of pure time preference, the consumption elasticity of marginal utility, and the country-specific growth rate, respectively, comprise the standard calibration of Nordhaus's DICE model[46] ($\rho = 1.5\%$, $\eta = 1.45$), the discounting assumed in the Stern review[47] ($\rho = 0.1\%$, $\eta = 1.01$), and the main specification by Ricke et al.[40] ($\rho = 2\%$, $\eta = 1.5$). |
| Probabilistic analysis | Historical growth response of countries to TCs (**UD4**) | 1200 bootstraps per country generated with a maximum entropy bootstrap method[43] respecting the spatial dependence structure of the original panel data. |
| | Response of TC impacts to future greenhouse gas emissions (**UD5**) | TC impact emulator[31] employed to generate 400 probabilistic time series of TCs with landfall, i.e., 100 time series for each of the 4 underlying global climate models. |
| | Response of temperature to a greenhouse gas emission pulse (**UD6**) | For each of the 4 underlying global climate models, and each of 15 carbon-cycle models, the response of global mean temperature to an additional emission pulse over the course of 100 years[48], as required in the SCC calculation. |

temperature effect at the 10% level of significance and vice versa (Methods). This suggests that TC impacts are indeed an additional impact channel not captured by population-weighted temperature over land alone (Fig. S1 and Methods). This is plausible because TC activity is rather linked to sea-surface temperature than to air temperature over land, and the number and locations of TCs with landfall – and thus the national annual shares of people exposed – are strongly influenced by confounding factors (e.g., atmospheric circulation conditions such as shear winds, location of genesis)[42]. Second, we employ a maximum entropy bootstrapping method[43] to express the uncertainties associated with the historical growth estimates. This approach also allows us to propagate the uncertainty to future projections of TC damages. The bootstrapping preserves spatio-temporal correlation patterns of the predictor resulting from TCs affecting more than one country. We generate 1200 bootstraps for each country and estimate the growth response individually for each bootstrap (Methods).

We find that TC strikes reduce growth in the short-, mid-, and long-term. Cumulative (median) growth losses first monotonously increase with the number of lag years considered before they somewhat remain constant for 6–13 lags (Fig. 1c). This highlights the importance to account for the accumulation of persistent growth impacts also in the future projection of TC damages. In the remainder of this paper, we use a model with eight lag years as *main specification* since it represents the best compromise between capturing the full persistence of damages and statistical robustness (Methods). The heterogeneous exposure of countries to TCs leads to substantially different induced growth losses (Figs. 1b, S3, and S5–S11). For instance, we find that the average annual growth losses for strongly exposed countries such as Japan (1.63%) or the Philippines (2.62%) were in the historical period on average 10 and 15 times higher than for the only partially exposed USA (0.17%).

**Uncertainty dimensions of future damage projections**
Future projections of climate damages are subject to substantial and compounding climate and socioeconomic uncertainties[44]. For the projections of TC damages in terms of DAD and TC-induced SCC, we identify six relevant uncertainty dimensions (UD) which we map out carefully by a combination of scenario analysis and probabilistic methods such as bootstrapping (Table 1).

The first and the second dimensions cover the dependency of our estimates on assumptions on future emissions and socioeconomic development. We here span a broad range of different emission

futures by accounting for three RCPs[32] and account for dependencies regarding future population and economic development by two different SSPs[34].

With the third uncertainty dimension, we capture the impact of normative assumptions on how the global society weighs future compared to present TC damages. These are expressed by discount rates $\rho + \eta g$ that are adjusted by the economic growth rate $g$ of the considered country[45]. In order to cover a broad range of discounting assumptions, we assess three standard choices for the rate of pure time preference $\rho$ and the consumption elasticity of marginal utility $\eta$ in the literature (as in the standard calibration of Nordhaus's DICE model[46], in the Stern Review[47], and as used by Ricke et al.[40]). Increasing $\rho$ reduces the relative weight with which society rates future compared to present damages while increasing $\eta$ enhances the willingness of society to further down-weight future damages if they foster economic and thus consumption growth today.

The fourth, fifth and sixth dimensions cover structural modeling uncertainties that occur along each of the different scenario combinations, i.e. uncertainty dimensions 1–3. The fourth dimension consists of uncertainties regarding the historical growth response of countries to TCs which is captured by bootstrapping. With the fifth dimension, we cover uncertainties regarding the changes of TC impacts (e.g., with respect to their location, timing, intensity, and frequency distributions of landfall) under future greenhouse gas (GHG) emissions. To this end, we make use of the statistical TC emulator introduced in ref. 31 that generates probabilistic time series of TCs based on the output from four different global climate models. We use the emulator to generate 100 probabilistic time series of TCs with landfall for each ocean basin where TCs occur and separately for each of the four underlying global climate models. These simulations are performed for each of the three RCPs and a "no further climate change" baseline scenario. The sixth dimension captures uncertainties regarding the response of global mean temperature to the emission of an additional ton of $CO_2$ over the course of 100 years[48].

Uncertainty dimensions 4–6 are fully propagated throughout the modeling chain to account for compounding uncertainties, and results are averaged only at the very end of the modeling chain and separately for each of the RCP-SSP-discounting scenario combinations.

**Discounted annual damage by tropical cyclones**
To evaluate how TCs will affect TC-prone countries under global warming, we here estimate the DAD over the period 2010–2100 as the

discounted difference of the national GDP trajectories with and without additional warming. We use the "no further climate change" baseline scenario as a reference to compare additional effects of global warming to present levels (Methods). To this end, we first calculate the wind field along each synthetic TC track and superimpose it with gridded population projections of the SSPs, to obtain time series of national shares of affected people. We choose 2010 as the start year ensuring that all projections start from the same initial population distributions and GDP levels. For each RCP-SSP-combination, TC realization, and bootstrap, we then calculate the resulting time series of national shares of affected people (e.g., see Fig. S4a, b for the USA under SSP2 and SSP5, respectively). Assuming that TCs in the future have the same long-term impacts on economic growth as in the historical period, we employ the empirical growth model to translate the shares of affected people into deviations of the national GDP pathways as provided by the SSPs (by construction, climate impacts are not accounted for in the SSPs[33]). Propagating all uncertainty dimensions to the very end, this amounts to 2,880,000 perturbed GDP time series for each of the 41 TC-affected countries (e.g., see Fig. S4e, f for the USA under SSP2 and SSP5, respectively). In the following, we will refer to the combination of RCP6.0-SSP2 and Ricke's discounting choice (cf. Table 1) as the *main specification*.

The global sum of the DAD is obtained by summing the DADs across all countries. The median global DAD averaged over the future period 2010–2100 and across uncertainty dimensions 4 and 5 (cf. Table 1) is positive across all RCP-SSP-discounting scenarios and ranges from 0.18% of global GDP in 2021 for the main specification up to 1.13% for RCP8.5-SSP5 and Stern's discounting choice (Fig. 2a and Tbl. S1). Discounting represents the primary source of uncertainty (uncertainty dimension 3 in Table 1); across all six RCP-SSP combinations, the median average global DAD for the lowest discount rate (Stern) is by a factor four higher than for the highest discount rate (Ricke). Uncertainties with regard to changes of the predictor (national shares of affected people) along the RCP pathways relative to the "no further climate change"-baseline scenario for each of the four GCMs (uncertainty dimension 5 in Table 1) are the 2[nd] largest source of uncertainty, followed by uncertainties with regard to the historical growth response to TC strikes (uncertainty dimension 4 in Table 1). Due to the effect of discounting, the variation of DAD across the 91 years of the averaging period (2010–2100) is comparably low and converges after 2070 (Figs. 2a and S14).

In absolute terms, the median average DAD per-capita (DADpc) of the six most strongly exposed high-income countries, ranging from US $ 170 (USA) to 2046 (Taiwan), is substantially higher than for the Philippines (US$ 139) and Mauritius (US$ 216) as the most affected lower-middle-income country and upper-middle-income country, respectively (Fig. 2b and Table S2). However, this difference between the income groups shrinks when DAD are measured relative to 2019 Average Household Income (AHI). For instance, the relative DADpc of the upper-middle-income island states of Mauritius (0.71% of AHI) and Jamaica (0.33% of AHI) as well as of the lower-middle-income countries of the Philippines (0.93% of AHI) and Vietnam (0.29% of AHI) are higher than for the USA (0.21% of AHI). This corresponds to 1.22, 3.39, 1.07 days of income lost per year for an average household in Jamaica, the Philippines, and Vietnam compared to 0.82 days of income lost per year for an average U.S. household. The DADpc strongly increase for lower values of the discount rate. For instance, the days of income lost per year for an average household in the Philippines increase from 3.39 to 12.21 under RCP6.0-SSP2 and Stern's low choice of the discount rate (Fig. S13).

## Country-level temperature-dependent damage functions

Climate integrated assessment models that are used to calculate optimal mitigation pathways in the presence of climate damages typically lack a detailed climate model to keep computational

expenses manageable. Instead, they employ reduced form relations[37], or simple climate models (e.g., MAGICC[49]) to calculate the response of GMT to GHG emissions. Temperature damage functions are then used to translate warming levels into economic damages, which allows to weigh the costs of mitigation measures with the avoided damages along endogenously calculated, non-standard (i.e., non-RCP) emission pathways. These functions are usually highly aggregated and lack the ability to resolve climate extremes and their impacts rigorously[50]. By contrast, we here derive event-based, country-level temperature-dependent damage functions for TCs that account for the persistence of damages in the economic system, and employ them to assess the contribution of TCs to the SCC.

A priori, it is not certain whether TC-induced growth losses can be expressed as time-independent functions of GMT for two main reasons. First, while GMT is closely related to other variables, such as boundary layer moisture content or sea level rise, changes in the frequency and intensity of TCs with landfall also depend upon changes in other meteorological variables such as atmospheric circulation changes (e.g. shear winds)[11]. This could render damages dependent upon the warming trajectories and thus the underlying RCP scenarios. On the other hand, changes in socioeconomic variables such as population development differ between the SSP scenarios, and this could result in an SSP dependence of damages. We use F-tests to test for structural dependencies of the relationship between GMT and TC-induced growth rate changes upon the underlying RCP and SSP scenarios (Methods). We find that the relationship is independent of the SSP scenarios at the 10% level of significance. This may be surprising at first because for different SSPs the TCs with landfall affect rather different national populations. However, the within-country population distributions are so similar that the national shares of affected people (the damage predictors) are very similar for both SSPs (cf. Fig. S4a, b for the USA under SSP2 and SSP5, respectively); for 90% (95%) of the national shares of people exposed for a specific RCP-year-country combination, the difference between the SSPs is less than 5% (10%). By contrast, the relationship depends (weakly) upon the RCPs, globally (Fig. 3a) as well as at the country-level (Figs. S14 and S15). We account for these dependencies as random effects in our regression analysis using a mixed-effects modeling approach (Methods) (see Fig. 3a for a visualization on the global level, and Figs. S14 and S15 for the country-level regressions). The comparably weak RCP dependence is inherited from the weak dependence of the national shares of affected people upon the RCPs (cf. Fig. S4c, d for the example of the USA under SSP2 and SSP5, respectively).

For 37 out of the 41 TC-affected countries considered, we find growth losses to robustly increase with GMT (Fig. 3b, confidence intervals established from uncertainty dimensions 4 and 5, Methods). We find similar shares of significantly negatively affected countries across all four income groups (high-income countries 91% (10 out of 11), upper-middle-income countries 85% (11 out of 13), lower-middle-income countries 93% (14 out of 15), and low-income countries 100% (2 out of 2)). This clearly illustrates that growth losses do not depend on the development level but rather on country-specific characteristics. Thus, also growth in high-income countries is negatively affected by TCs suggesting that, even in this country group, historically implemented adaptation measures may not suffice to compensate for climate change-induced loss increases in the future.

## Tropical cyclone-induced social cost of carbon

We next employ the temperature-dependent damage functions to estimate the SCC induced by TCs. To this end, we use an approach introduced by Ricke et al.[40] which evaluates the difference in future GDP per-capita projections under climate damage with and without an additional emission pulse (Methods). This allows us to estimate the SCC at the country-level as well as globally, where the global SCC is defined as the sum of the country-level contributions. To estimate the

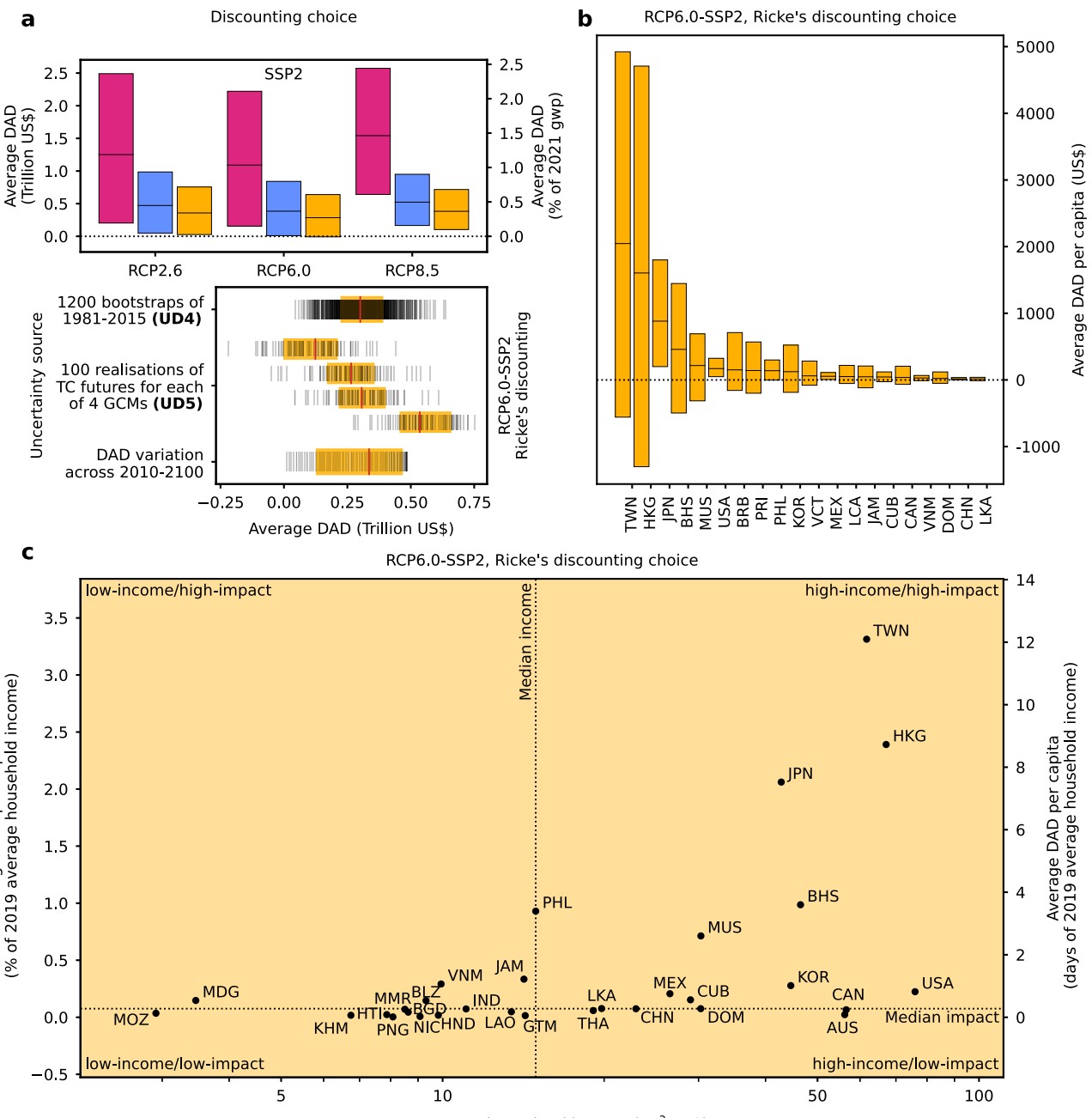

**Fig. 2 | Discounted annual damage by tropical cyclones. a** Upper panel: Median global discounted annual damage (DAD) from tropical cyclone impacts averaged over the period 2010–2100 for three Representative Concentration Pathways (RCP2.6, 6.0, 8.5), two Shared Socioeconomic Pathways (SSP2 and SSP5), and three different values of the growth adjusted discount rates used in the Stern review[47], for the standard calibration of Nordhaus's DICE model[46], and by Ricke et al.[40] (color code, Table 1) in Trillion US\$ (left y-axis) and relative to global world product (GWP, right y-axis). Black horizontal lines, and bounds of boxes indicate median losses and 66% confidence interval (17–83%) accounting for uncertainty dimensions (**UDs**) 4 and 5 (n = 120, 000 realizations (Table 1)). Lower panel: Quantification of uncertainties for the main specification (RCP6.0-SSP2, and Ricke's discounting choice). The uncertainty drivers listed on the y-axis are allowed to vary, all other dimensions of uncertainties are averaged out (Table 1). Each vertical line is a point estimate. Red lines, and bounds of shaded boxes denote the median and 66% confidence interval (17–83%) across each of the listed sources of uncertainty, respectively. **b** Median country-level per-capita DAD averaged over the period 2010–2100 for the 20 most impacted countries and the main specification (RCP6.0-SSP2, and Ricke's discounting choice). Black whiskers indicate the 66% confidence interval (17–83%) accounting for the uncertainty dimensions 4 and 5 (n = 120, 000 realizations per country (Table 1)). **c** Median average per-capita DAD for the main specification relative to 2019 average household income (left y-axis); note log-scale on x-axis. The right y-axis shows median per-capita DAD in terms of days of average household income lost. The quadrants classify countries by above (high) and below (low) median income and per-capita DAD across the exposed countries for which income data is available. See supplementary Tbl. S2 for country codes in **b** and **c.**

SCC, Ricke et al.[40] use the statistical relationship between per-capita GDP growth and population-weighted annual mean temperatures over land derived by Burke et al.[41]. For consistency reasons and to avoid double counting, we here combine Burke's temperature terms with terms measuring the TC impacts in a common regression framework

(Methods). Calculating the SCC once with and once without TC impacts and subtracting the latter from the former allows us to estimate the absolute TC-induced SCC (TC-SCC) as well as the relative increase in SCC due to TCs. For our main specification and without TC impacts, we find a median global SCC of US\$/tCO$_2$ 173 (2005 constant

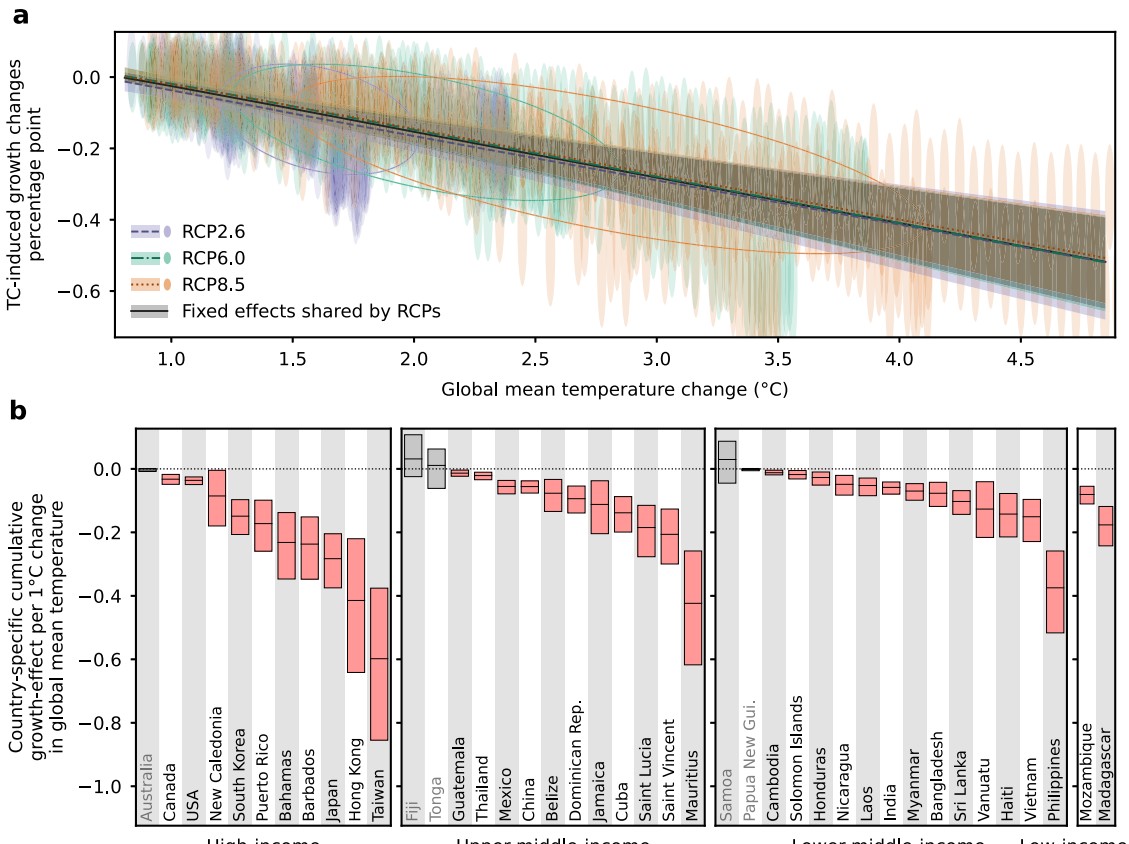

**Fig. 3 | Temperature-dependent damage functions for tropical cyclone induced growth losses. a** Visualization of the quasi-universal scaling of global (country-averaged) tropical cyclone-induced growth losses with global mean temperature (GMT) change (relative to pre-industrial levels) across Representative Concentration Pathways (RCPs) 2.6 (blue), 6.0 (orange), and 8.5 (red). Large ellipses and markers denote the one-standard-deviation confidence range and the 66% confidence range of annual relative growth losses for each RCP across uncertainty dimensions 2, 4, and 5 (n=240, 000 realizations (Table 1)). The lines and shaded areas illustrate the result of the mixed-effects model fit: The black line is the fixed effect shared by all RCPs while the colored lines include the RCP-specific random effects. Shaded areas indicate 66% confidence intervals. Since the average across countries is unweighted, the figure provides only a visualization of the relative RCP and SSP independence of the damage functions; for quantitative estimates refer to country-level damage functions (Figs. S14 and S15 and **b**). **b** Country-level growth rate changes per degree of GMT change for high-income, upper-middle-income, lower-middle-income, and low-income countries (World Bank's 2024 income classification scale (https://datahelpdesk.worldbank.org/knowledgebase/articles/906519)). Black horizontal lines and bounds of boxes indicate median losses and 66% confidence intervals as obtained from uncertainty dimensions 4 and 5 ($n = 120, 000$ realizations per country (Table 1)), respectively, and red and gray colors denote statistically significant and non-significant results, respectively. Coefficients of country-level damage functions are provided in Tbl. S3. Parameters: 8 lag years and Ricke's discounting choice (main specification).

US\$) (66% confidence interval (CI): US\$/tCO$_2$ 108–266) (Tbls. S4 and S5). When accounting for TCs, median SCC increases substantially by 22% to US\$/tCO$_2$ 212 (138–318). Not surprisingly, the global SCC is highest for the strongest emission scenario RCP8.5 for both SSP scenarios (e.g., US\$/tCO$_2$ 361 (222–588) vs. US\$/tCO$_2$ 212 (138–318) for the main specification). As for the DAD, we find that the discount rate has the largest impact on the SCC estimates across all six RCP-SSP scenario combinations; for the lowest discount rate (Stern) the global SCC (including TC effects and for SSP2-RCP6.0) is US\$/tCO$_2$ 1654 (1122–2361), more than seven times higher than for the highest discount rate (our main specification).

The TC-SCC of a country is determined by its exposure and economic vulnerability to TCs (as described by the damage functions) but also by its economic output. Since the latter is closely linked to the country's GHG emissions[51], only strongly affected large emitters have notable TC-induced increases in SCC (Fig. 4b). In absolute terms, the USA perceives the largest increase in median SCC through TCs from US\$/tCO$_2$ 13 (66% CI: US\$/tCO$_2$ 3–26) to 20 (8–35), representing 17.7% of the median global TC-SCC, followed by Japan (from US\$/tCO$_2$ 1 (0–2) to 8 (5–12), 17.6% of global median TC-SCC), Taiwan (from US\$/tCO$_2$ 2 (2–3) to 8 (5–12), 14.4% of global median TC-SCC), China (from US

\$/tCO$_2$ 8 (3–16) to 14 (7–23), 14.4% of global median TC-SCC), and India (from US\$/tCO$_2$ 43 (30–64) to 47 (33–70), 11.3% of global median TC-SCC). While median country-level TC-SCC is moderate (e.g., US\$/tCO$_2$ 6.64 for the USA), it is subject to relatively large uncertainties, and there are high tail risks (Fig. 4b). For instance, the 66% CI (established from uncertainty dimensions 4–6) for the USA spans from US\$/tCO$_2$ 4.34 to 9.79 (Fig. 4c).

## Discussion

We combine process-based modeling of the population exposure to TCs with an empirical growth model into a transparent, tractable, and openly accessible framework for assessing the future socioeconomic impacts of these events. We apply the framework to TCs but it could be extended to other categories of extreme weather events for which event-based projections of impact indicators are available. These are, for instance, provided by impact model inter-comparison projects[52].

With DAD and TC-SCC, we employ two complementary metrics to quantify the additional burdens that climate change imposes on national economies and societies through TC impacts. For both metrics, we show the importance of empirically constraining the persistence of growth losses in the economic system and accounting for

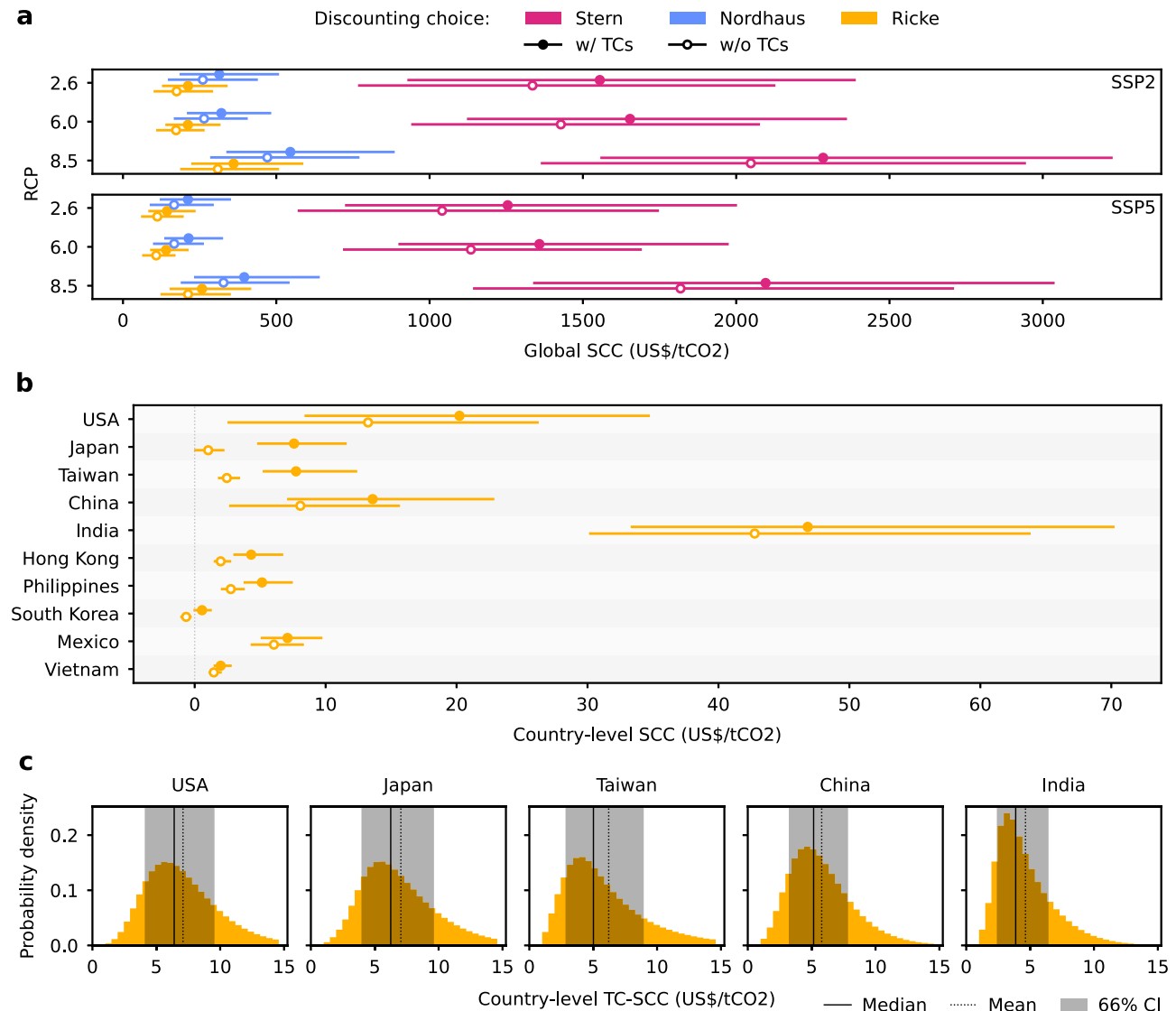

**Fig. 4 | Tropical cyclone-induced social cost of carbon. a** Global median social cost of carbon (SCC) for the period 2010–2100 with and without TC effects, for three Representative Concentration Pathways (RCP2.6, 6.0, 8.5), two Shared Socioeconomic Pathways (SSPs 2 and 5), and three different values of the growth-adjusted discount rates used in the Stern review[47], for the standard calibration of Nordhaus's DICE model[46], and by Ricke et al.[40] (color code, Table 1). Error bars indicate the 66% confidence interval (17–83%) accounting for the uncertainty dimensions 4–6 ($n = 7,200,000$ realizations (Table 1)). **b** Median country-level SCC with and without TC effects. Error bars indicate the 66% confidence interval (17–83%) accounting for the uncertainty dimensions 4–6 ($n = 7,200,000$ realizations per country (Table 1)). **c** Probability distributions of the country-level TC-SCC across uncertainty dimensions 4–6 for Ricke's choice of the discount rate ($n = 7,200,000$ realizations per country (Table 1)). For these countries, 100% of the TC-SCC values are positive. Parameters: 8 lags, RCP6.0-SSP2, and Ricke's discounting choice (main specification), except in **a**.

them in the estimates of TC damages (Figs. S5–S11). Our DAD analysis reveals that differences in income among countries must not be neglected when estimating and comparing the adaptation challenges of countries. In per-capita terms, absolute DADs are highest for strongly exposed high-income countries such as Taiwan, Japan, and the USA. However, when DAD is measured relative to average household income, the damages for the strongly exposed, upper-middle-income island states of Mauritius and Jamaica but also for the lower-middle-income countries of the Philippines and Vietnam become comparable to those of the USA. For an average household, these range from one to more than 10 days of income lost per year, strongly depending upon assumptions on discounting (cf. Fig. S13a). Since low- and middle-income countries often have less financial means to support their domestic population and invest in reconstruction efforts in the disaster aftermath[53], our estimates suggest that, in relative terms, TC-prone low- and middle-income countries could face even larger

adaptation challenges than their high-income counterparts. Further, our DAD analysis likely underestimates the economic implications for lower-income populations that are often more exposed to – and need longer to recover from – TC strikes[54].

There is a large number of SCC estimates in the literature. A recent meta-analysis including 207 studies and more than 5,000 SCC estimates finds that median SCC estimates have quadrupled over the last decade from US$/tCO$_2$ 48 in 2007–2012 to US$/tCO$_2$ 328 in 2018–2022 for Ricke's choice of the rate of pure time preference ($\rho = 2\%$, our main specification)[55]. This increase reflects our growing knowledge on climate impacts. Our preferred estimate is with US$/tCO$_2$ 212 (66% CI: US$/tCO$_2$ 138–318) somewhat lower than the median estimate of the meta-analysis for 2018–2022. This is not surprising since we only resolve the impacts from annual temperatures and TCs, explicitly. While our approach may still account implicitly for other impact channels such as health impacts[56], several

other channels are neglected completely. For instance, we do not account for climate change impacts on natural capital[57], and we do not value the risk of catastrophic events resulting e.g. from the transgression of climate tipping points[58,59]. Our SCC estimates can most directly be compared to the estimate of Ricke et al. [40] since we use their methodology to derive the SCC. They report a somewhat higher median SCC of US\$/tCO$_2$ 417 (95% CI: US\$/tCO$_2$ 161–732). The main reason is that our estimate for the temperature effects differs – though not significantly at 5% level – from the estimate by Burke et al.[41] on which Ricke et al.[40] base their SCC estimates. Most studies entering the meta-analysis do not explicitly account for TC impacts, with the exception of studies based on the IAM FUND[15,60]. For instance, as one of the older estimates accounted for in the meta-analysis, the FUND-based study of Narita et al.[15] reports a comparably low overall SCC of US\$/tCO$_2$ 2.5 ($\rho = 1\%$). The authors estimate TCs to contribute only about 1% to the overall SCC. The relatively small contribution of TCs to the overall SCC may result from two factors. First, the use of a TC damage function that assumes a simple scaling of damages with GMT instead of estimating future damages from synthetic TC tracks as done in more recent studies by Mendelsohn et al.[35] and Bakkensen et al.[17] as well as in this study. Second, Narita et al.[15] do not account for the persistence of TC damages in the economic system. Further, a number of studies entering the meta-analysis account for TC contributions to the SCC indirectly through damages to other sectors as for instance agriculture[61]. Further, several SCC estimates are based on the damage function used in Nordhaus's DICE model[37]. It is obtained from a meta-analysis of impact studies and adds a 25% adjustment to account for various categories of damages that are difficult to monetize such as extreme weather events but also biodiversity loss, ocean acidification, and catastrophic events. The substantial increase in SCC by 22.2% at the global level and by 44.4% in TC-affected countries that we observe when explicitly accounting for TC impacts and their persistence in the economic system, suggests that a more explicit representation of TCs could drive up previous SCC estimates that do not, or only indirectly, account for TC impacts. An inclusion of TC impacts in other assessment frameworks may be facilitated by the temperature damage functions presented in this study.

In our study, we assume that countries' growth response to future TCs mimics their historical response. This may result in an over-estimation of damages when future adaptation measures reduce the vulnerability of countries to TCs, or in an underestimation of damages in case of maladaptation. For instance, previous empirical works did not find evidence of adaptation to TCs in the USA[16,29]. Furthermore, our damage estimates are based solely on the wind fields of the storms, and we neglect TC-related rainfall[17] and storm surge[62] as drivers of TC damage. Since the storm-surge risk increases with rising sea levels, and also TC rainfall is projected to intensify under global warming[11,63], our damage projections should be considered as a lower bound of the overall expected damage from TCs.

Our modeling approach lacks the flexibility of fully-fledged climate integrated assessment modeling frameworks[64]; climate damages in terms of DAD and SCC can be calculated along standard climate- and socio-economic scenarios but the framework does not allow us to endogenously model mitigation responses to climate damages (e.g., investment in renewable energies, carbon taxation, etc.) as well as their impact on global warming and on the socioeconomic development of countries. However, by deriving empirical country-level temperature dependent-damage functions accounting for the persistence of damages in the economic system, we pave the way for the integrated assessment modeling community to perform such fully integrated analyses.

Further, it is critical to keep in mind that these estimates only comprise the contributions of TCs, which affect a limited set of countries. Other major categories of extreme weather events such as floods and droughts cause direct damages that are comparable to TC damages globally (annual economic losses of US\$ 37.8bn and 8.5bn averaged over the last decade, compared to US\$ 51.5bn for TCs, according to ref. [1]) and also reduce the economic growth of the affected countries in the long term[2,65]. Thus, their contributions to i) the global SCC and ii) the SCCs of major emitters such as China, the USA, and India – the 1$^{st}$, 2$^{nd}$, 3$^{rd}$ largest emitters – that are not only prone to TCs but also to fluvial floods or droughts may be comparable to those of TCs. Our study suggests that the TC damages can be rather substantial already, driving up the country-level SCC of China, the USA, and India by 68%, 53%, and 9%, respectively. Since DAD and SCC estimates are key for decision- and policymakers at national and international levels charged with weighing the costs of mitigation and adaptation measures with climate change impacts and empower them to soundly design mitigation policies such as carbon taxation schemes[37], our findings underline the importance of adequately accounting for the contributions of all major categories of extreme weather events in future damage estimates.

## Methods

### Economic Data

For the historical period, we derive national economic growth rates for the period 1981–2015 from a harmonized data set of national time series of annual (constant 2005 US\$) purchasing power parity-adjusted (PPP) per-capita GDP (GDPpc) as provided by the Institute for Health Metrics and Evaluation of the University of Washington[66]. For the future projections, we employ the national GDPpc projections as provided in the IIASA *Basic Elements* SSP database (https://tntcat.iiasa.ac.at/SspDb/) as part of the *OECD Env-Growth* model in constant 2005 US\$ (PPP). Since the GDPpc projections of the SSPs are only given in five-year time steps, we interpolate them to annual resolution using cubic splines. For the comparison of temperature-dependent TC impacts on growth (cf. Fig. 3b), countries are grouped by the World Bank's 2024 income classification scale (https://datahelpdesk.worldbank.org/knowledgebase/articles/906519) into high-income (2022 per–capita gross national income (GNIpc) US\$≥13, 846), upper-middle-income (US\$4, 466≤GNIpc≤US\$13, 845)), lower-middle-income (US\$1, 136≤GNIpc≤US\$4, 465), and low-income countries (GNIpc≤US\$1, 135).

To compare estimates of the social cost of carbon (SCC) in the literature with our own estimates, we use the time series of GDP deflators for the USA provided by the World Bank (https://data.worldbank.org/indicator/NY.GDP.DEFL.ZS?locations=US) to express all SCC estimates in constant 2005 US\$.

### Tropical cyclone affected population

We consider the annual time series of the share of people exposed to TCs over the study period 1950–2015 provided as part of the Tropical Cyclone Exposure Database (TCE-DAT[67]). The share of TC-exposed people is estimated based on the wind footprint of each storm by summing the population of all grid-cells experiencing maximum (1-minute sustained) wind speeds of at least 34 knots. TCE-DAT uses population data from the HYDEv3.2.1 database[68] remapped to annual resolution and to the spatial resolution of the wind fields (6 arc-minutes ≈ 11 km at the equator) to derive the annual shares of national populations exposed to TCs; we consider the yearly sum of people exposed to TCs across all events.

For the TC projections, we use the TC track emulator of ref. [31]. It regresses frequency and intensity statistics of 614,400 synthetic TCs with landfall from the Inter-Sectoral Impact Model Intercomparison Project[52,69] (ISIMIP) with regard to GMT and Equatorial Southern Oscillation Index (ESOI) in the TC basins and sub-basins East Pacific, North Atlantic, North Indian, South Indian East, South Indian West, West Pacific North, West Pacific South, South Pacific West, and South Pacific East over the period 1861–2100 according to four CMIP5-GCMs

(HadGEM2-ES, MIROC5, IPSL-CM5A-LR and GFDL-ESM2M) participating in ISIMIP2b[52] and the RCPs 2.6, 6.0, and 8.5.

For each GCM-RCP combination and (sub-)basin, we generate 100 probabilistic time series of TC tracks for the period 1980–2100. In total, this amounts to 1200 artificial TC time series for each ocean basin. We contrast the TCs and their impacts for the different RCP-SSP combinations with a "no further climate change" baseline scenario. According to HadCRUT5 data[70], GMT for the period 2005–2020 ranged between 0.8 and 1.3 °C above its level for the pre-industrial period 1850–1900 (including the boundaries of 95% confidence intervals). In order to allow for larger probabilistic variability in our model, we use a slightly larger range of between 0.5 °C and 1.5 °C. For each GCM-RCP-region combination, we then compute the mean frequency and intensity of landfalling TCs in years where the global mean temperature falls within this interval. With that value of frequency and intensity held fixed over 1980–2100 we again generate 100 probabilistic realizations[31].

The impact assessment tool CLIMADA[71] is used to generate wind footprints for each TC track. The number of people exposed to these 1-minute sustained winds of at least 34 knots is aggregated by country and year. The population exposure for the future period is generated by extending the historical HYDEv3.2.1 population data set[68] for each SSP over the period 2010–2100[72]. The data is downscaled to 300-arcsec from the original 450-arcsec resolution and linearly merged to the historical data starting from 2005. Since the physical impacts have been calibrated on a TC basin level, the number of people per country affected by the simulated TCs exhibits considerable bias when compared to the numbers in TCE-DAT over the observational period 1950–2015 (see above). Therefore, we multiply the simulated time series of affected people per country by a correction factor so that the simulated average over 1980–2015 agrees with the respective historical country average according to TCE-DAT.

### Countries included in the analysis

Our analysis is applied to the set of 166 countries used by Burke et al.[41] and Ricke et al.[40], and an additional 8 TC-affected regions and countries for which economic data was available (Barbados, Hong Kong, Jamaica, St. Lucia, Myanmar, New Caledonia, Tonga, and Taiwan). In total, we identify 41 of the 174 countries to be affected by TCs in the historical period 1981–2015.

### Country-level average temperature time series

Our source of historical temperature data is the GSWP3-W5E5 historical reconstruction included in ISIMIP3a[73] (https://doi.org/10.48364/ISIMIP.982724.1), which is a back-extension of W5E5 v2.0[74] data from 1979-2019 to 1901-2019 using the Global Soil Wetness Project phase 3 (GSWP3) dataset[75]. To represent temperature effects in our panel model, we aggregate the 0.5 degree grid cell estimates to the country-year level, weighting by the annual population density according to the HYDEv3.3-based data for 1950–2020 provided within ISIMIP3a[73] (https://doi.org/10.48364/ISIMIP.822480.2) and using the fractional country mask that is provided as part of ISIMIP3a (https://doi.org/10.48364/ISIMIP.635131.2).

For the computation of SCC over the future period 2025–2100, we use the population-weighted data along GCMs and RCPs provided as part of the code repository of an earlier study on country-level SCC by Ricke et al.[40]. The data is available for all four GCMs considered in this study (HadGEM2-ES, MIROC5, IPSL-CM5A-LR and GFDL-ESM2M), and for three RCPs (RCP4.5, RCP6.0, and RCP8.5) for the period 2006–2105. For our analysis of RCP2.6, we use the RCP4.5 data since Ricke et al.[40] do not provide population weighted temperature data for RCP2.6. Furthermore, we bias-correct the temperature series relative to the historical ISIMIP3a data (see above), by shifting according to the difference in averages over the years 2006-2020.

### Global mean temperature time series

For the calibration of the TC emulator we use global mean temperature time series from CMIP5 according to four different GCMs (HadGEM2-ES, MIROC5, IPSL-CM5A-LR and GFDL-ESM2M), and three RCPs (RCP2.6, RCP6.0, and RCP8.5). A 21-year running mean smoothing and normalization relative to piControl are applied[31]. The same global mean temperature time series are used for the estimation of temperature-dependent TC damage functions and of the TC-related social cost of carbon (see below). All steps are applied to the GCM-specific time series first and aggregation is applied as late as possible.

### Regression approach

To estimate the national GDPpc growth responses over the period 1981–2015, we use a three-way fixed effects panel model,

$$g_{i,t} = \gamma_i + \delta_t + \theta_i^0 t + \theta_i^1 t^2 + \alpha^0 T_{i,t} + \alpha^1 T_{i,t}^2 + \sum_{l=0}^{L} \beta_l P_{i,t-l} + \epsilon_{i,t}. \quad (1)$$

Here, $g_{i,t} \equiv \ln(y_{i,t}) - \ln(y_{i,t-1})$ denotes the per-capita GDP growth rate of country $i$ at time $t$ that is calculated from the difference of the logarithms of the GDPpc, $y_{i,t}$, at time $t$ and $t-1$. Further, $\gamma_i$, $\delta_t$, $\theta_i^0$, $\theta_i^1$ are country and time fixed-effects and country-specific time trends, respectively, and $\epsilon_{i,t}$ denotes the error term. Following Burke et al.[41], the global non-linear effect of temperature is expressed as the quadratic function $\alpha^0 T_{i,t} + \alpha^1 T_{i,t}^2$ of the population-weighted average temperature indicator $T_{i,t}$ for country $i$ at time $t$; the coefficients $\alpha^0$ and $\alpha^1$ describe the average growth response across all countries. Finally, the sum $\sum_{l=0}^{L} \beta_l P_{i,t-l}$ describes the cumulative growth response of country $i$ to the national shares of people exposed to strong TC winds, $P_{i,t-l}$ in the year $t-l$, for $l \in [[0, ..., L]]$ lag years, where $L$ denotes the maximum number of lag-years; the coefficients $\beta_l$ describe the average growth response across all countries. We apply an ordinary least squares regression approach, and find that the results agree with a robust linear regression approach[2,76].

We implement a bootstrapping strategy to account for uncertainties related to the *economic growth regression* (Eq. (1)). As TCs can affect more than one country[3,4], our panel data may be spatially dependent. This is why we use a maximum entropy method[43] to construct 1200 bootstraps (*b*) respecting the dependence structure of the original panel data.

Considering the p-values (computed across 1200 bootstraps) of the coefficients $\beta_l, l = 0, ..., L$ for lag numbers between $L = 0$ and $L = 15$, we find that all coefficients for $7 \leq l \leq 13$ except $l = 8$ and $l = 11$ are non-significant at the 5% level of significance. Since higher lag numbers come with increasingly large uncertainty (Fig. S2), we choose $L = 8$ lag years as main specification.

To test the robustness of our regression approach, we study different versions of the regression model of Eq. (1) and compare the estimated temperature impact to the regression results of Burke et al.[41]. To this end, we use the same structure of the fixed effects and the same temperature indicators as Burke et al. but base our regressions on updated per-capita GDP and population-weighted temperature time series for a larger set of countries. We compare our preferred model that included TC effects to i) simpler versions of the model not accounting for TC effects or temperature effects and ii) Burke et al.'s regression for the period 1981–2015 (used throughout this study) as well as for the period 1961–2010 used by Burke et al. We find that the temperature effects estimated with the different models are not significantly different from the effects reported by Burke et al. (5% significance level, $p \geq 0.09$). Further, by omitting the TC impact terms, we find that these do not affect the temperature coefficients at the 10% level of significance ($p \geq 0.94$). Vice versa, no significant change in TC effects at the 10% level of significance ($p \geq 0.87$) is found when omitting the temperature effect in our model.

**Projection of discounted annual damages.** For each of the 41 TC-affected countries, $i$, three different Representative Concentration Pathways (RCPs) ($e \in$ {RCP2.6, 6.0, and 8.5}) and one "no further climate change" baseline scenario (bl), two Shared Socioeconomic Pathways (SSPs) ($s \in$ {SSP2 and SSP5}), four bias-corrected CMIP5-GCMs ($m \in$ { HadGEM2-ES, MIROC5, IPSL-CM5A-LR, and GFDL-ESM2M})[52], and 100 probabilistic TC time series, $r$, for each RCP-SSP-GCM scenario combinations, we project a time series of TC-induced changes to the national GDPpc growth rates over the period $t \in [[2010, 2100]]$,

$$\phi_{i,t}^{e/\text{bl},s,m,r,b,l} \equiv \sum_{l'=0}^{l} \beta_{l'}^{b,l} P_{i,t-l'}^{e/\text{bl},s,m,r,b}. \tag{2}$$

For each country, we obtain 3,801,600 RCP/baseline($e$/bl)-SSP($s$)-GCM($m$)-TC time series($r$)-bootstrap($b$) scenario combinations. Further, to assess the impact of the persistence of growth losses, the growth rate changes are calculated for $l = 8$ lag years.

From these growth rate changes, we calculate the perturbed national GDPpc pathways for each of the SSPs as

$$y_{i,t}^{e/\text{bl},s,m,r,b,l} \equiv y_{i,t-1}^{e/\text{bl},s,m,r,b,l} e^{g_{i,t}^s + \phi_{i,t}^{e/\text{bl},s,m,r,b,l}}, \tag{3}$$
$$\text{for } t \in [[2010,\dots,2100]] \text{ and } y_{i,2010}^{e/\text{bl},s,m,r,b,l} = y_{i,2010}^s,$$

where $y_{i,2010}^{e/\text{bl},s,m,r,b,l}$ denotes the initial national GDP of country $i$ for the year 2010, and $g_{i,t}^s = \ln(y_{i,t}^s) - \ln(y_{i,t-1}^s)$ is the country $i$'s GDP growth rate at year $t$ as derived from SSP scenario $s$, respectively.

For each scenario combination and maximum lag time, the DAD of country $i$ are then defined as the discounted difference between the GDP of the "no further climate change" (bl) scenario and the corresponding scenarios for the different RCPs,

$$\text{DAD}_{i,t}^{e,s,m,r,b,l} \equiv \left( y_{i,t}^{\text{bl},s,m,r,b,l} - y_{i,t}^{e,s,m,r,b,l} \right) \prod_{\tau=2010}^{t} e^{-r_{i,\tau}^{\text{bl}}}, \tag{4}$$
$$\text{for } t \in [[2010,\dots,2100]],$$

where $r_{i,\tau}^{\text{bl}} = \rho + \eta g_{i,t}^{\text{bl}}$ denotes the growth-adjusted discount rate with the rate of time preference $\rho$ and the consumption elasticity of marginal utility $\eta$, relative to the per-capita GDP growth rate $g_{i,t}^{\text{bl}}$ at year $t$ as derived from the baseline scenario bl.

### Estimation of temperature-dependent damage functions
We derive temperature-dependent damage functions by linearly regressing the growth rate changes $\Delta\phi_{i,t}^{e,s,m,r,b,l} \equiv \phi_{i,t}^{e,s,m,r,b,l} - \phi_{i,t}^{\text{bl},s,m,r,b,l}$ relative to the baseline with the change in GMT in year $t$, $\Delta T_t^{e,m}$, with regard to pre-industrial levels, using the mixed-effects model

$$\Delta\phi_{i,t}^{e,s,m,r,b,l} \equiv (\overline{\gamma}_i^{r,b,l} + \gamma_i^{e,r,b,l}) + (\overline{\omega}_i^{r,b,l} + \omega_i^{e,r,b,l})\Delta T_t^{e,m} + \epsilon_i^{r,b,l}, \tag{5}$$

where the fixed effects parameters $\overline{\gamma}_i^{r,b,l}$ and $\overline{\omega}_i^{r,b,l}$ are shared by all RCPs, while the random effects parameters $\gamma_i^{e,r,b,l}$, and $\omega_i^{e,r,b,l}$ follow a bivariate distribution with mean zero to account for distributional differences between the RCPs. Note that the fit is performed across all GCM-SSP scenarios combinations which is why these coefficients are GCM- and SSP-independent. In Fig. 3b and in the SCC computations (below), only the fixed effects parameters $\overline{\gamma}_i^{r,b,l}$ and $\overline{\omega}_i^{r,b,l}$ are used.

To test for structural dependencies of the TC induced growth losses upon SSP and RCP scenarios, we run two OLS regressions, where the first regression includes a coefficient that does not depend on the SSPs, and the second regression includes a coefficient that does not depend on the RCPs,

$$\Delta\phi_{i,t}^{e,s,m,r,b,l} \equiv (\overline{\gamma}_i^{r,b,l} + \gamma_i^{e,r,b,l}) + (\overline{\omega}_i^{r,b,l} + \omega_i^{e,r,b,l})\Delta T_t^{e,m} + \epsilon_i^{e,r,b,l}, \tag{6a}$$

$$\Delta\phi_{i,t}^{e,s,m,r,b,l} \equiv (\overline{\gamma}_i^{r,b,l} + \gamma_i^{s,r,b,l}) + (\overline{\omega}_i^{r,b,l} + \omega_i^{s,r,b,l})\Delta T_t^{e,m} + \epsilon_i^{s,r,b,l}. \tag{6b}$$

### Structural dependence tests
We then test for structural breaks in the intercept and slope coefficients across SSPs and RCPs by running F-tests for the hypotheses $\gamma_i^{s,r,b,l} = 0$, $\omega_i^{s,r,b,l} = 0$, and $\gamma_i^{e,r,b,l} = 0$, $\omega_i^{e,r,b,l} = 0$, respectively. We find that the hypotheses $\gamma_i^{e,r,b,l} = 0$ and $\omega_i^{e,r,b,l} = 0$ have to be rejected at the 5% level of significance for between 58.5% and 86.5% of the samples across uncertainty dimensions 5 and 6 ($r$ and $b$), depending on the country $i$. Thus, the relationship between the exogenous and endogenous variables cannot be explained by the RCP-independent coefficients $\overline{\gamma}_i^{r,b,l}$ and $\overline{\omega}_i^{r,b,l}$ alone, and we therefore account for the structural dependence on RCPs by including the random variables $\gamma_i^{e,r,b,l}$ and $\omega_i^{e,r,b,l}$. By contrast, the hypotheses $\gamma_i^{s,r,b,l} = 0$ and $\omega_i^{s,r,b,l} = 0$ cannot be rejected at the 10% level of significance ($p \geq 0.69$) for any country $i$ and for any of the samples across uncertainty dimensions 5 and 6 ($r$ and $b$), which is the reason why we omit these variables in the regression of Eq. (5).

### Calculation of the social cost of carbon from tropical cyclones
To compute the SCC, we follow the approach by Ricke et al.[40]. The SCC is the additional damage incurred by an additional emission pulse, for which we chose 1 gigatonne of carbon (GtC) in 2025. The corresponding temperature response to the pulse is approximated using the coupled carbon-climate convolution integral approach from ref. 48, according to 15 carbon-cycle models combined with the four GCMs that we used in the TC projections (HadGEM2-ES, MIROC5, IPSL-CM5A-LR and GFDL-ESM2M). Empirically estimated temperature effects are used to project the temperature-induced changes in the national per-capita GDP pathways, as given by the SSPs for each RCP-GCM combination until 2100, with and without the additional emission pulse in 2025.

To account for the direct effects of temperature on the national level, we follow Ricke et al.[40] by using the increase in the empirically estimated aggregate temperature effect $\alpha^0 T_{i,t} + \alpha^1 T_{i,t}^2$ relative to 2025, where $T_{i,t}$ is the national population-weighted annual mean temperature for country $i$ in year $t$. However, while Ricke et al.[40] use the coefficients from ref. 41, we use our own estimates of $\alpha^0$ and $\alpha^1$ (Eq. (1)). To project the TC-induced changes driven by global mean temperature change, we add the full TC-realizations-bootstrap ensemble of TC damage functions (Eq. (5)) to the aforementioned temperature effects.

The additional effect of the pulse for country $i$ and for a given climate change scenario (cc) is then given by

$$D_{i,t}^{\text{pulse}} \equiv \begin{cases} \dfrac{-(y_{i,t}^{\text{cc}+\text{pulse}} - y_{i,t}^{\text{cc}})p_{i,t}}{1GtC} & \text{if } t \leq 2100, \\ D_{i,2100}^{\text{pulse}} & \text{else} \end{cases} \tag{7}$$

with $p_{i,t}$ denoting the population of country $i$. The time horizon for the SCC calculation is 2200, and we assume constant additional effects $D_{i,t}^{\text{pulse}}$ after 2100. The country-level SCC is finally calculated as the net present value of the additional damage from the pulse, applying growth-adjusted discounting as described above,

$$\text{SCC}_{i,2025} \equiv \sum_{t=2025}^{2200} D_{i,t}^{\text{pulse}} \prod_{\tau=2025}^{t} e^{-r_{i,\tau}^{\text{cc}}}. \tag{8}$$

The per-capita GDP growth rate $g_{i,t}^{\text{cc}} \equiv \ln(y_{i,t}^{\text{cc}}) - \ln(y_{i,t-1}^{\text{cc}})$ used in the discount rate $r_{i,\tau}^{\text{cc}} = \rho + \eta g_{i,\tau}^{\text{cc}}$ is derived from the per-capita GDP pathway without the additional emission pulse. For years after 2100, the growth rate $g_{i,t}^{\text{cc}} \equiv \max(g_{i,2100}^{\text{cc}}, 0)$ is assumed to be constant with the value of the year 2100 fixed, with negative growth rates truncated to 0 (as in ref. 48). Note that, to be consistent with the calculation of the DAD, we here slightly deviate from Ricke's implementation[48] where the

discrete time definition of discount rates is used. Summing over all countries then yields the global SCC.

Further, to calculate the SCC without the explicit contribution of TCs, we repeat the same procedure omitting the temperature-dependent damage function for TCs, and only using the empirically estimated aggregate temperature-dependent damage function. The TC-induced SCC (TC-SCC) is then defined as the difference between the SCC with and without the contribution of TCs.

### Reporting summary

Further information on research design is available in the Nature Portfolio Reporting Summary linked to this article.

## Data availability

Machine-readable source data for all figures, supplementary figures, and supplementary tables are provided with this paper as CSV files. The GDP, population, temperature, and tropical cyclone (TC) exposure data sets that support the findings of this study are available from https://doi.org/10.5281/zenodo.8063450. National annual time series of historical per-capita GDP (GDPpc) are openly provided by the Institute for Health Metrics and Evaluation of the University of Washington[66]. The national GDPpc projections are provided by the IIASA *Basic Elements* SSP database (https://tntcat.iiasa.ac.at/SspDb/) as part of the *OECD Env-Growth* model in constant 2005 US$ (PPP). The World Bank's 2024 income classification scale is openly accessible through (https://datahelpdesk.worldbank.org/knowledgebase/articles/906519). The source of historical temperature data is the GSWP3-W5E5 historical reconstruction (https://doi.org/10.48364/ISIMIP.982724.1) which are weighted by annual, HYDEv3.3-based population densities (https://doi.org/10.48364/ISIMIP.822480.2) and using the fractional country mask (https://doi.org/10.48364/ISIMIP.635131.2) included in ISIMIP3a[73]. The historical TC exposure data is available from the Tropical Cyclone Exposure Database (TCE-DAT)[67] using population data from the HYDEv3.2.1 database[68]. The population projections according to the SSPs used for the projections of affected people are openly available from ref. 72. The TC track simulations that were used to generate the probabilistic future TC exposure data sets are available for scientific purposes only and upon request from WindRiskTech (info@windrisktech.com). The TC emulator additionally employs daily temperature data from four CMIP5 GCMs (Had-GEM2-ES, MIROC5, IPSL-CM5A-LR and GFDL-ESM2M) and three RCPs (RCP4.5, RCP6.0, and RCP8.5) for the future period 2006–2105 as provided within ISIMIP2b and openly accessible through the ISIMIP data portal (https://data.isimip.org/search/tree/ISIMIP3a/tree/ISIMIP2b/InputData/climate/atmosphere/). The temperature level for the "no further climate change" baseline scenario is based on the HadCRUT5[70] global mean temperature time series that is openly accessible at the Met Office Hadley Centre (https://www.metoffice.gov.uk/hadobs/hadcrut5/data/current/download.html). For the SCC analysis, population-weighted temperature data along the four CMIP5 GCMs and three RCPs are provided as part of the code repositories of Ricke et al.[40] (https://github.com/country-level-scc/cscc-paper-2018) and Burke et al.[41] (https://purl.stanford.edu/wb587wt4560). The country shapes in Figs. 1a and S12b are available in the public domain from the Natural Earth project website (https://www.naturalearthdata.com/downloads/110m-cultural-vectors/110m-admin-0-countries/). The per-capita income in Fig. 2c is openly accessible as part of the World Inequality Database[77] (https://wid.world/bulk_download/wid_all_data.zip). Source data are provided with this paper.

## Code availability

The model to generate the probabilistic TC tracks from climate model outputs is intellectual property of WindRiskTech (info@windrisktech.com) and cannot be shared publicly. All remaining code that was used i) for the regression analyses, ii) the generation of the future

tropical cyclone (TC) exposure indicators, and iii) the calculation of damage estimates, and iv) to analyze the data and produce the figures was implemented in Python 3.9 (https://www.python.org/) with CLI-MADA 3.3.3 (https://zenodo.org/record/7691855) and statsmodels 0.13.5 (https://www.statsmodels.org/), and is openly available from https://doi.org/10.5281/zenodo.8056520.

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

## Acknowledgements

This research has received funding from the German Federal Ministry of Education and Research (BMBF) under the research projects SLICE (01LA1829A), and QUIDIC (01LP1907A) and from the CHIPS project, part of AXIS, an ERA-NET initiated by JPI Climate, funded by FORMAS (Sweden), DLR/BMBF (Germany, grant no. 01LS1904A), AEI (Spain) and ANR (France) with co-funding by the European Union (grant no. 776608).

## Author contributions

H.K., C.O., and T.V. designed the method. T.G. and T.V. provided the TC exposure. C.S. generated the historical population-weighted temperature time series. F.P. calculated the SCC. H.K. and T.V. conducted the rest of the simulations. All authors contributed to the analysis, and H.K and C.O. wrote the manuscript with contributions from all authors. All authors discussed the results.

## FundingInformation

## Competing interests

The authors declare no competing interests.
