## [Peer Review File · Nature Communications]

The Social Cost of Tropical CyclonesREVIEWER COMMENTS

Reviewer #1 (Remarks to the Author):

Reviewer report

The Social Cost of Tropical Cyclones

Krichene, Vogt, Piontek, Geiger, and Otto develop country-specific, temperature-dependent tropical cyclone damage functions and show that both DAD and SCC increase when TC are explicitly accounted for. While I consider it is an interesting paper, there are some issues that should be clarified and previous literature on the topic needs to be cited, and results should be compared and discussed.

These are my main concerns:

- The authors do not acknowledge TC damage functions have been previously developed and estimates of the contribution of TC to total damages have been reported (see annotated pdf).
 - TC damages are incorporated implicitly (and in some cases explicitly) in IAMs damage functions. As such SCC estimates do include the influence of TC, at least to some extent. Note that even if TC may not be treated in a separate category, damages from TC in different sectors are included (crop damage or losses due to increased flooding, winds or other effects of TC). This is not acknowledged in the paper.
 - While the empirical damage function from Burke et al. does not include TC explicitly, it is very likely that TC damages are included in it implicitly. The authors do not show that this is not the case, or modify the damage function to exclude such damages. The result is that this may generate double counting problems.
 - Estimates of the SCC-TC should be compared with previous SCC that include explicitly or implicitly TC. This would help to better gauge the importance of the contribution of this paper.
 - The authors assume that their estimated damage functions are SSP and RCP independent. However, this is based on visual inspection of a graph and not formally tested (see annotated pdf).
- Some minor comments are included in the attached annotated pdf.

Reviewer #2 (Remarks to the Author):

The authors derived novel temperature-dependent damage functions for 41 tropical cyclone-affected countries to quantify the country-level social cost of carbon induced by the persistent growth effects of damaging cyclones. The methodology use is sound, detailed and appropriate. Their results show that accounting for 13 tropical cyclone impacts significantly increases the social cost carbon of the strongly affected greenhouse gas emitters. This work is original and very significant to the climate change discourse and other related fields. It supports the claims and conclusions arrived at.

Response to Reviews

Reviewer #1:

The Social Cost of Tropical Cyclones

Krichene, Vogt, Piontek, Geiger, and Otto develop country-specific, temperature-dependent tropical cyclone damage functions and show that both DAD and SCC increase when TC are explicitly accounted for. While I consider it an interesting paper, there are some issues that should be clarified and previous literature on the topic needs to be cited, and results should be compared and discussed.

These are my main concerns:

1. The authors do not acknowledge TC damage functions have been previously developed and estimates of the contribution of TC to total damages have been reported (see annotated pdf).

We thank the reviewer for this constructive feedback. Before, we focused on contributions to the literature accounting for potential long-term effects of tropical cyclones on output growth. We would very much agree with the reviewer that this might leave the reader with the wrong impression that there are only very few works considering TC damages in general. In the revised version of the manuscript (MS), we now give a much more comprehensive overview over the literature. The corresponding paragraph of the *Introduction* reads:

Over the years, a number of TC damage functions were developed, ranging from rather simple estimations of how damages change with global mean temperature (GMT) and socioeconomic development (Pielke 2007; Narita, Tol, and Anthoff 2009) to more complex event-based approaches accounting for characteristics of individual storms such as areas affected by strong TC winds (Geiger, Frieler, and Levermann 2016) and lifetime rainfall (Bakkensen, Park, and Sarkar 2018). Many works are USA specific (Watson and Johnson 2004; Schmidt, Kemfert, and Höppe 2010; Nordhaus 2010; Emanuel 2011; Murnane and Elsner, n.d.; Zhai and Jiang 2014) or employ damage functions derived for the USA to estimate damages in other countries (Sealy and Strobl 2017; Elliott, Strobl, and Sun 2015). Considerably fewer studies derive damage functions accounting for the TC climatology and socioeconomic vulnerabilities of other TC-prone countries such as the Philippines (Strobl 2019; Baldwin et al. 2023), China (Bakkensen, Shi, and Zurita 2018), or South-Korea (Bakkensen, Park, and Sarkar 2018), or to derive global sets of region-specific damage functions (Bakkensen and Mendelsohn 2016; Eberenz, Lüthi, and Bresch 2021).

With the notable exceptions of (Hsiang and Jina 2014) and (Elliott, Strobl, and Sun 2015), projections of future damages in the literature are based on damage functions which statistically link TC predictors to reported (direct) damages. This approach has the advantage that damage databases can be employed for calibration, but it does not allow to account for the persistence of damages in the economic system. This neglect could result in an underestimation of (future) damages and limit the usefulness of these earlier estimates for national policymakers and actors of international climate finance charged with estimating adaptation needs at the country level and comparing them across countries.

[...]

Similarly to earlier works by (Mendelsohn et al. 2012) and (Bakkensen, Park, and Sarkar 2018), we employ large sets of synthetic TCs with landfall to account for changes in TC climatology with global warming. In addition, we here account for the persistence of damages in the economic system, building on earlier works by (Elliott, Strobl, and Sun 2015) and (Hsiang and Jina 2014). We finally express damages as functions of GMT change. These temperature damage functions are needed by most climate integrated assessment models to

weigh the costs of mitigation and adaptation measures with the remaining climate change impacts (Narita, Tol, and Anthoff 2009).

2. TC damages are incorporated implicitly (and in some cases explicitly) in IAMs damage functions. As such SCC estimates do include the influence of TC, at least to some extent. Note that even if TC may not be treated in a separate category, damages from TC in different sectors are included (crop damage or losses due to increased flooding, winds or other effects of TC). This is not acknowledged in the paper.

We thank the reviewer for this constructive comment, and we would agree that the formulation in the submitted version was misleading. In the revised MS, we now acknowledge that for instance studies based on the integrated assessment model FUND explicitly account for TC damages and that they are indirectly included through other damage categories in different integrated assessment modeling frameworks. Please see our answer to this reviewer's comment #4 for a reproduction of the corresponding paragraph in the *Discussion* of the revised MS.

3. While the empirical damage function from Burke et al. does not include TC explicitly, it is very likely that TC damages are included in it implicitly. The authors do not show that this is not the case, or modify the damage function to exclude such damages. The result is that this may generate double counting problems.

We thank the reviewer for this remark that encouraged us to revise the design of our panel growth regressions. We now estimate the impacts of tropical cyclones and annual temperatures (using the same linear and quadratic temperature terms and fixed effects as (Burke, Hsiang, and Miguel 2015)) in a common regression framework. We then use our estimates for the impacts of temperature on economic activity to estimate the SCC without TC contributions instead of the Burke damage functions that we used in the previous version of the manuscript. We would consider this approach more consistent for two reasons. First, it assures that damages from tropical cyclones and temperature are estimated from the exact same per-capita GDP data and for the same time period. Second, it minimizes the risk of double counting. We test whether temperature and TC effects are independent by comparing the full regression model to versions of the model where either the TC variables or the temperature variables are omitted. We find that the coefficients of the temperature variables do not change significantly (at the 10% level of significance) when the TC terms are omitted and vice versa, suggesting that these are independent impact channels. We would consider this also plausible from a biophysical point of view. First, because TC activity is rather linked to sea-surfaces temperatures than to (population-weighted) air temperatures over land (used as temperature predictor). Second, the number and locations of TCs with landfall – and thus the national annual shares of people exposed (TC impact predictor) – are strongly influenced by confounding factors (e.g., atmospheric circulation conditions such as shear winds, location of genesis) (Knutson et al. 2019).

The corresponding paragraph in Section *Country-level temperature-dependent damage functions* of the *Results* now reads:

We add methodologically to this previous work in two regards. First, we additionally control for the impact of temperature on growth. Thereby, we follow the approach of (Burke, Hsiang, and Miguel 2015) and account for linear and quadratic terms in population-weighted annual mean temperatures over land for a set of 174 countries that includes the 41 TC-affected countries. Estimating temperature and TC impacts on growth in a single regression framework allows also to investigate potential interactions of both effects. If temperature and TCs can be considered as independent impact channels, their contributions to growth losses become additive.

We find that including TC effects does not affect the temperature effect at the 10% level of significance and vice versa (Methods). This suggests that TC impacts are indeed an additional impact channel not captured by population-weighted temperature over land alone (Fig. S1 and Methods).

This is plausible because TC activity is rather linked to sea-surface temperature than to air temperature over land, and the number and locations of TCs with landfall – and thus the national annual shares of people

exposed – are strongly influenced by confounding factors (e.g., atmospheric circulation conditions such as shear winds, location of genesis) (Knutson et al. 2019).

Second, [...]

More details on the regression approach are provided in the Methods and supplementary Figure S1.

4. Estimates of the SCC-TC should be compared with previous SCC that include explicitly or implicitly TC. This would help to better gauge the importance of the contribution of this paper.

We would agree with the reviewer that a comparison with the SCC estimates in the literature helps to judge the validity of our approach and to assess the importance of the contribution of TCs to the overall SCC. At the same time this comparison is challenging since there is a wealth of SCC estimates, obtained with very different modeling approaches, out there. Referring to all of them is not possible within the limit of 70 references. Therefore, we decided to compare our results, on the one hand, to a recent meta-analysis by (Tol 2023) including 207 studies with over 5000 estimates. We further directly compare our results to those by (Ricke et al. 2018) since we use their methodology to estimate the “baseline” SCC (not accounting for the SCC contributions). The corresponding section in the Discussion reads:

There is a large number of SCC estimates in the literature. A recent meta-analysis including 207 studies and more than 5,000 SCC estimates finds that median SCC estimates have quadrupled over the last decade from US\$/tCO₂ 48 in 2007–2012 to US\$/tCO₂ 328 in 2018–2022 for Ricke's choice of the rate of pure time preference ($\rho=2\%$, our main specification) (Tol 2023). This increase reflects our growing knowledge on climate impacts. Our preferred estimate is with US\$/tCO₂ 212 (66% CI: US\$/tCO₂ 138–318) somewhat lower than the median estimate of the meta-analysis for 2018–2022. This is not surprising since we only resolve the impacts from annual temperatures and TCs, explicitly. While our approach may still account implicitly for other impact channels such as health impacts (Rennert et al. 2022), several other channels are neglected completely. For instance, we do not account for climate change impacts on natural capital (Bastien-Olvera and Moore 2020), and we do not value the risk of catastrophic events resulting e.g. from the transgression of climate tipping points (Cai and Lontzek 2019; Dietz et al. 2021). Our SCC estimates can most directly be compared to the estimate of (Ricke et al. 2018) since we use their methodology to derive the SCC. They report a somewhat higher median SCC of US\$/tCO₂ 417 (95% CI: US\$/tCO₂ 161–732). The main reason is that our estimate for the temperature effects differs – though not significantly at 5% level – from the estimate by (Burke, Hsiang, and Miguel 2015) on which (Ricke et al. 2018) base their SCC estimates.

Most studies entering the meta-analysis do not explicitly account for TC impacts, with the exception of studies based on the IAM FUND (Narita, Tol, and Anthoff 2009; Anthoff and Tol 2013). For instance, as one of the older estimates accounted for in the meta-analysis, the FUND-based study of (Narita, Tol, and Anthoff 2009; Anthoff and Tol 2013) reports a comparably low overall SCC of US\$/tCO₂ 2.5 ($\rho=1\%$). The authors estimate TCs to contribute only about 1% to the overall SCC. The relatively small contribution of TCs to the overall SCC may result from two factors. First, the use of a TC damage function that assumes a simple scaling of damages with GMT instead of estimating future damages from synthetic TC tracks as done in more recent studies by (Mendelsohn et al. 2012) and (Bakkensen, Park, and Sarkar 2018) as well as in this study. Second, (Narita, Tol, and Anthoff 2009) do not account for the persistence of TC damages in the economic system.

Further, a number of studies entering the meta-analysis account for TC contributions to the SCC indirectly through damages to other sectors as for instance agriculture (Moore et al. 2017). Further, several SCC estimates are based on the damage function used in Nordhaus's DICE model (Nordhaus 2017). It is obtained from a meta-analysis of impact studies and adds a 25% adjustment to account for various categories of damages that are difficult to monetize such as extreme weather events but also biodiversity loss, ocean acidification, and catastrophic events.

This leads us to the conclusion that

The substantial increase in SCC by 22.2% at the global level and by 44.4% in TC-affected countries that we observe when explicitly accounting for TC impacts and their persistence in the economic system, suggests that a more explicit representation of TCs could drive up previous SCC estimates that do not, or only indirectly, account for TC impacts. An inclusion of TC impacts in other assessment frameworks may be facilitated by the novel temperature-damage functions presented in this study.

5. The authors assume that their estimated damage functions are SSP and RCP independent. However, this is based on visual inspection of a graph and not formally tested (see annotated pdf).

We thank the author for this comment that encouraged us to also reconsider our regression approach for the temperature damage functions for TCs. We now apply F-tests to test for potential structural dependencies of the TC-induced growth rate changes upon i) SSPs and ii) RCPs. We find that the growth rate changes are independent of the SSP scenarios at the 10% level of significance. Which we explain with the similarity of the within-country population distributions across the two SSPs.

By contrast, we find that the growth rate changes depend (weakly) upon the RCPs. In the revised version of the manuscript, we use a mixed-effects modeling approach (instead of ordinary least square regressions) which allows us to model RCP dependencies as random effects.

The corresponding paragraphs in Section *Country-level temperature-dependent damage functions* of the Results now reads:

We use F-tests to test for structural dependencies of the relationship between GMT and TC-induced growth rate changes upon the underlying RCP and SSP scenarios (Methods). We find that the relationship is independent of the SSP scenarios at the 10% level of significance. This may be surprising at first because for different SSPs the TCs with landfall affect rather different national populations. However, the within-country population distributions are so similar that the national shares of affected people (the damage predictors) are very similar for both SSPs (cf. Figs. S4a and S4b for the USA under SSP2 and SSP5, respectively); for 90% (95%) of the national shares of people exposed for a specific RCP-year-country combination, the difference between the SSPs is less than 5% (10%). By contrast, the relationship depends (weakly) upon the RCPs, globally (Fig. 3a) as well as at the country-level (Figs. S12 and S13). We account for these dependencies as random effects in our regression analysis using a mixed-effects modeling approach (Methods) (see Fig. 3a for a visualization on the global level, and Figs. S12 and S13 for the country-level regressions). The comparably weak RCP dependence is inherited from the weak dependence of the national shares of affected people upon the RCPs (cf. Figs. S4c and S4d) for the example of the USA under SSP2 and SSP5, respectively).

We would further like to mention that we also improved the visualization in Fig. 3a to better reflect the RCP-dependence. In the submitted version of the manuscript, markers denoted annual relative growth losses averaged across uncertainty dimensions 4 and 5 and the 41 TC-affected countries (as noted in the caption). Since each marker only represented an average, the distribution underlying each marker (which also influenced the regression result) was hidden. In particular, the differences between climate models that are covered by uncertainty dimension 5 were not visible. Instead, we now use an ellipsoidal marker shape that allows us to represent not only the average, but also the variability of the underlying distribution. Furthermore, we use individual markers for data points from different climate models (part of uncertainty dimension 5).

Minor comments

P2, 34-35: This sentence is weirdly structured.

We rephrased this sentence.

P2, 36: Please add some examples

We now provide a broad overview over the different approaches in the literature to project TC damages in the Introduction (cf. our answer to this reviewer's comment #1). The corresponding section of the Introduction now reads

P3, 50: The following paper should be cited and discussed as it is closely related to the present article: The impact of climate change on global tropical cyclone damage. Mendelsohn, Emanuel, Chonabayashi and Bakkensen

We thank the reviewer for pointing us towards this important study. We now discuss and link it to our work. The corresponding section in the Introduction now reads:

Similarly to earlier works by (Mendelsohn et al. 2012) and (Bakkensen, Park, and Sarkar 2018), we employ large sets of synthetic TCs with landfall to account for changes in TC climatology with global warming. In addition, we here account for the persistence of damages in the economic system, building on earlier works (Elliott, Strobl, and Sun 2015) and (Hsiang and Jina 2014). We finally express damages as functions of GMT change. These temperature damage functions are needed by most climate integrated assessment models to weigh the costs of mitigation and adaptation measures with the remaining climate change impacts (Narita, Tol, and Anthoff 2009).

P4, 86-89: This sentence is a bit convoluted.

We have rephrased this sentence. It now reads:

The heterogeneous exposure of countries to TCs leads to substantially different induced growth losses (Fig. 1b, Fig. S3, and Figs. S5–S9). For instance, we find that the average annual growth losses for strongly exposed countries such as Japan (1.63%) or the Philippines (2.62%) were in the historical period on average 10 and 15 times higher than for the only partially exposed USA (0.17%).

P5, 137: What is the starting year? 2021? Depending on it, there may be non-trivial differences in the initial values of population across the different SSPs. This could create a bias. The same can happen with GDP.

In the submitted version of the MS, we have indeed started the projection in 2021. Following the reviewer's suggestion, we now start in 2010 where the underlying population and GDPpc data for both SSPs are identical. In our case, we did not observe any relevant changes in the discounted annual damages from this shift in the start year.

P7, 169-174: This sentence is too long and not clear.

We have amended our language as explained in our response to the next remark.

P7, 173: "several days" is not informative.

We would agree, and have reformulated the sentence as:

In absolute terms, the median average DAD per-capita (DADpc) of the six most strongly exposed high-income countries, ranging from US\$ 170 (USA) to 2,046 (Taiwan), is substantially higher than for the two most affected low- and middle-income countries Philippines (US\$ 139) and Jamaica (US\$ 47) (Fig. 2b and Tab. S2). However, this difference between the income groups shrinks when DAD are measured relative to 2019 Average Household Income (AHI). For instance, the relative DADpc of the Small Island Developing State of Jamaica (0.33% of AHI) as well as of the lower-middle income countries of the Philippines (0.93% of AHI) and Vietnam (0.29% of AHI) are higher than for the USA (0.21% of AHI), corresponding to 1.22, 3.39, 1.07 days of income lost per year for an average household in Jamaica, the Philippines, and Vietnam compared to 0.82 days of income lost per year for an average U.S. household. The DADpc strongly increase for lower values of the discount rate. For instance, the days of income lost per year for an average household in the Philippines increase from 3.39 to 12.21 under RCP6.0-SSP2 and Stern's low choice of the discount rate (Fig. S11).

P8, 186-187: Why? This is not necessarily true. There are damage functions that incorporate persistence, and that depend on the rate of warming. These are dependent of the emissions pathway.

We would agree, and it was not a-priori clear whether it would be possible to express our event-based damage estimates along the different (but fixed) RCP-SSP pathway combinations. Our motivation for trying it is twofold: First, it allows us to employ Ricke et al.'s pulsing technique to calculate the SCC (Ricke et al. 2018). Second, many integrated assessment models include climate damages through temperature damage functions. Therefore, deriving those functions may allow to include TC impacts and their persistence into these models. However, we would agree that the formulation in the submitted version of the manuscript can be misleading. Therefore, we have left it out.

P8, 189: SSTs are one example for which pattern scaling techniques have been proven to be adequate (for most emissions scenarios). This means that they can be approximated by GMT multiplied by a constant. This is not a good example of what you want to argue.

We would agree and now use a more precise wording (following (Knutson et al. 2020)). The corresponding paragraph in the Section *Country-level temperature-dependent damage functions* now reads:

A priori, it is not certain whether TC-induced growth losses can be expressed as time-independent functions of GMT for two main reasons. First, while GMT is closely related to other variables, such as boundary layer moisture content or sea level rise, changes in the frequency and intensity of TCs with landfall also depend upon changes in other meteorological variables such as atmospheric circulation changes (e.g. shear winds) (Knutson et al. 2020).

P8, 193-196: The evaluation is based on looking at the graph and is highly subjective. I am not convinced that the estimated functions are RCP independent. The statistical similarity of the slope parameters of individual damage functions should be evaluated with a test. Moreover, the parameter stability of the slope parameter of the regression should be tested with a structural change test.

We thank the reviewer for this constructive comment that motivated us to revise our regression approach. Indeed, employing F-tests we find a dependence of the TC-induced growth losses upon the RCPs. We now employ a mixed effects approach instead of OLS regression which allows to account for the RCP dependencies as random effects. Unfortunately, in the submitted version of the MS, we chose a potentially misleading visualization where we only showed data points obtained by aggregation over bootstraps, climate models and TC realizations. We have changed this in the amended version of the figure.

P8, 199-204: see previous comment

See answer above.

P8, 206: No objective evidence is provided to support this claim.

As discussed in our response to the major comment #5 of this reviewer, we now use F-tests to test whether the TC induced growth losses are structurally independent of the underlying SSP.

P9, 222: Not accurate. Some IAMs do include TC. It should be clarified that TC are included in several damage functions under categories labeled as "extremes, disasters" or more commonly as a factor behind damages in sectors (e.g., agriculture). See, for example, the review about the costs of climate change by Tol (2019) and Tol, Narita and Anthoff (2008) in particular for an explicit inclusion of TC in FUND.

As discussed in our response to the major comment #4 of this reviewer, we now elaborate on how TC effects are (indirectly or directly) included in previous integrated assessment exercises.

P9, 235: It would be useful to provide confidence intervals.

We provide confidence intervals in the amended version of the MS.

P10, 240-241: Not clear why this assumption is expected to hold. The estimates are empirical and they very likely include the effects of TC in an indirect manner, if not explicitly. Double counting is likely.

We thank the reviewer for this interesting and constructive remark. Wondering how we could avoid double counting, we have revised our regression approach. To this end, we included the temperature terms of the Burke regression into our regression framework to obtain more consistent estimates of the contribution of annual temperature and TCs to overall damages. As also discussed in our response to the major comment #3 by this reviewer, we find that the temperature parabola does not change significantly (at 10% level) when the TC terms are added and vice versa. We would consider this as a strong indication that annual population-weighted temperatures over land national shares of people affected by strong TC winds are indeed independent impact channels. For clarity, we have changed our language such that we mainly discuss increases in the SCC (as obtained from the temperature terms) when explicitly including the TC terms. These increases should be robust, even if the predictors are correlated to a certain degree.

P10, 248-251: It would be useful to report confidence intervals. Do the intervals include negative values?

We now include the 66% confidence intervals to all SCC values mentioned in the text. For the countries discussed in the text all SCC values are positive. However, there are several countries for which the SCC is negative. For instance, it can be seen from Fig. 4b that the median SCC is negative for South Korea when TC effects are not included.

P18, 427: They cannot be identical and in very good approximation at the same time. Moreover, as mentioned in a previous comment this is not shown to be the case. Only a visual comparison is presented.

We thank the reviewer for pointing us to this lapsus. We have amended our wording in the corresponding section of the Methods. Further, as discussed in our response to the reviewer's major remark #5, we now apply F-tests to assess potential dependencies of our TC-induced growth rate changes (damages) upon the underlying RCP and SSP scenarios. We find that the TC-induced growth losses depend upon the RCP scenarios. We now use a mixed-effect model instead of ordinary least square regressions which allows to account for the RCP dependencies as random effects.

P18, 437-438: Burke's damage function is very likely to include implicitly damages from TC.

As discussed in our response to the major comment #3 of this reviewer, we have extended our panel regression approach by including terms that are linear and quadratic in population weighted annual national mean temperatures (i.e., the same as in the regression approach used by (Burke, Hsiang, and Miguel 2015).

This allows us to avoid double counting and to obtain more consistent estimates of temperature and TC effects. We find that the coefficients of the temperature terms do not change significantly (10% confidence level) when the TC terms are omitted and vice versa, suggesting that these are independent impact channels. We would consider this also plausible from a biophysical point of view. First, because TC activity is rather linked to sea-surfaces temperatures than to population weighted air temperatures over land (used as temperature predictor). Second, the number and locations of TCs with landfall – and thus the national annual shares of people exposed (TC impact predictor) – are strongly influenced by confounding factors (e.g., atmospheric circulation conditions such as shear winds, location of genesis) (Knutson et al. 2019).

Reviewer #2:

The authors derived novel temperature-dependent damage functions for 41 tropical cyclone-affected countries to quantify the country-level social cost of carbon induced by the persistent growth effects of damaging cyclones. The methodology use is sound, detailed and appropriate. Their results show that accounting for 13 tropical cyclone impacts significantly increases the social cost carbon of the strongly affected greenhouse gas emitters. This work is original and very significant to the climate change discourse and other related fields. It supports the claims and conclusions arrived at.

We thank the reviewer for this very encouraging assessment of our work. In response to the comments made by reviewer #1, we have refined our regression approach in the sense that we now estimate the impacts of tropical cyclones and annual temperature (using the same linear and quadratic temperature terms and fixed effects as (Burke, Hsiang, and Miguel 2015)) in a common regression framework. We then use our estimates for the impacts of temperature on economic activity to estimate the SCC without TC contributions (instead of the Burke damage functions that we used in the previous version of the manuscript). We would consider this approach more consistent for two reasons. First, it assures that damages and temperature are estimated for the exact same GDP data and the same time period. Second, it minimizes the risk that damages are double counted due to potential dependence of TC affected people on annual temperature. We find that the coefficients of the temperature terms do not change significantly (10% confidence level) when the TC terms are omitted and vice versa, suggesting that these are independent impact channels. We would consider this also plausible from a biophysical point of view. First, because TC activity is rather linked to sea-surfaces temperatures than to population weighted air temperatures over land (used as temperature predictor). Second, the number and locations of TCs with landfall – and thus the national annual shares of people exposed (TC impact predictor) – are strongly influenced by confounding factors (e.g., atmospheric circulation conditions such as shear winds, location of genesis) (Knutson et al. 2019).

References

- Anthoff, David, and Richard S. J. Tol. 2013. "The Uncertainty about the Social Cost of Carbon: A Decomposition Analysis Using Fund." *Climatic Change* 117 (3): 515–30.
- Bakkensen, Laura A., and Robert O. Mendelsohn. 2016. "Risk and Adaptation: Evidence from Global Hurricane Damages and Fatalities." *Journal of the Association of Environmental and Resource Economists* 3 (3): 555–87.
- Bakkensen, Laura A., Doo-Sun R. Park, and Raja Shanti Ranjan Sarkar. 2018. "Climate Costs of Tropical Cyclone Losses Also Depend on Rain." *Environmental Research Letters: ERL [Web Site]* 13 (7): 074034.
- Bakkensen, Laura A., Xiangying Shi, and Brianna D. Zurita. 2018. "The Impact of Disaster Data on Estimating Damage Determinants and Climate Costs." *Economics of Disasters and Climate Change* 2 (1): 49–71.
- Baldwin, Jane W., Chia-Ying Lee, Brian J. Walsh, Suzana J. Camargo, and Adam H. Sobel. 2023. "Vulnerability in a Tropical Cyclone Risk Model: Philippines Case Study." *Weather, Climate, and Society* -1 (aop). <https://doi.org/10.1175/WCAS-D-22-0049.1>.
- Bastien-Olvera, Bernardo A., and Frances C. Moore. 2020. "Use and Non-Use Value of Nature and the Social Cost of Carbon." *Nature Sustainability* 4 (2): 101–8.
- Burke, Marshall, Solomon M. Hsiang, and Edward Miguel. 2015. "Global Non-Linear Effect of Temperature on Economic Production." *Nature* 527 (7577): 235–39.
- Cai, Yongyang, and Thomas S. Lontzek. 2019. "The Social Cost of Carbon with Economic and Climate Risks." *The Journal of Political Economy* 127 (6): 2684–2734.
- Dietz, Simon, James Rising, Thomas Stoerk, and Gernot Wagner. 2021. "Economic Impacts of Tipping Points in the Climate System." *Proceedings of the National Academy of Sciences of the United States of America* 118 (34). <https://doi.org/10.1073/pnas.2103081118>.
- Eberenz, Samuel, Samuel Lüthi, and David N. Bresch. 2021. "Regional Tropical Cyclone Impact Functions for Globally Consistent Risk Assessments." *Natural Hazards and Earth System Sciences* 21 (1):

393–415.

- Elliott, Robert J. R., Eric Strobl, and Puyang Sun. 2015. "The Local Impact of Typhoons on Economic Activity in China: A View from Outer Space." *Journal of Urban Economics* 88 (July): 50–66.
- Emanuel, Kerry. 2011. "Global Warming Effects on U.S. Hurricane Damage." *Weather, Climate, and Society* 3 (4): 261–68.
- Geiger, Tobias, Katja Frieler, and Anders Levermann. 2016. "High-Income Does Not Protect against Hurricane Losses." *Environmental Research Letters: ERL [Web Site]* 11 (8): 084012.
- Hsiang, Solomon M., and Amir S. Jina. 2014. "The Causal Effect of Environmental Catastrophe on Long-Run Economic Growth: Evidence From 6,700 Cyclones." Working Paper Series. National Bureau of Economic Research. <https://doi.org/10.3386/w20352>.
- Knutson, Thomas, Suzana J. Camargo, Johnny C. L. Chan, Kerry Emanuel, Chang-Hoi Ho, James Kossin, Mrutyunjay Mohapatra, et al. 2019. "Tropical Cyclones and Climate Change Assessment: Part I: Detection and Attribution." *Bulletin of the American Meteorological Society* 100 (10): 1987–2007.
- . 2020. "Tropical Cyclones and Climate Change Assessment: Part II: Projected Response to Anthropogenic Warming." *Bulletin of the American Meteorological Society* 101 (3): E303–22.
- Mendelsohn, Robert, Kerry Emanuel, Shun Chonabayashi, and Laura Bakkensen. 2012. "The Impact of Climate Change on Global Tropical Cyclone Damage." *Nature Climate Change* 2 (3): 205–9.
- Moore, Frances C., Uris Baldos, Thomas Hertel, and Delavane Diaz. 2017. "New Science of Climate Change Impacts on Agriculture Implies Higher Social Cost of Carbon." *Nature Communications* 8 (1): 1607.
- Murnane, R. J., and J. B. Elsner. n.d. "Maximum Wind Speeds and US Hurricane Losses." *Geophysical Research Letters*. <https://doi.org/10.1029/2012GL052740>.
- Narita, D., R. S. J. Tol, and D. Anthoff. 2009. "Damage Costs of Climate Change through Intensification of Tropical Cyclone Activities: An Application of FUND." *Climate Research* 39 (June): 87–97.
- Nordhaus, William D. 2010. "THE ECONOMICS OF HURRICANES AND IMPLICATIONS OF GLOBAL WARMING." *Climate Change Economics* 01 (01): 1–20.
- . 2017. "Revisiting the Social Cost of Carbon." *Proceedings of the National Academy of Sciences of the United States of America* 114 (7): 1518–23.
- Pielke, Roger A., Jr. 2007. "Future Economic Damage from Tropical Cyclones: Sensitivities to Societal and Climate Changes." *Philosophical Transactions. Series A, Mathematical, Physical, and Engineering Sciences* 365 (1860): 2717–29.
- Rennert, Kevin, Frank Errickson, Brian C. Prest, Lisa Rennels, Richard G. Newell, William Pizer, Cora Kingdon, et al. 2022. "Comprehensive Evidence Implies a Higher Social Cost of CO₂." *Nature* 610 (7933): 687–92.
- Ricke, Katharine, Laurent Drouet, Ken Caldeira, and Massimo Tavoni. 2018. "Country-Level Social Cost of Carbon." *Nature Climate Change* 8 (10): 895–900.
- Schmidt, Silvio, Claudia Kemfert, and Peter Höpfe. 2010. "The Impact of Socio-Economics and Climate Change on Tropical Cyclone Losses in the USA." *Regional Environmental Change* 10 (1): 13–26.
- Sealy, Kathleen Sullivan, and Eric Strobl. 2017. "A Hurricane Loss Risk Assessment of Coastal Properties in the Caribbean: Evidence from the Bahamas." *Ocean & Coastal Management* 149 (November): 42–51.
- Strobl, Eric. 2019. *The Impact of Typhoons on Economic Activity in the Philippines: Evidence from Nightlight Intensity*. Asian Development Bank.
- Tol, Richard S. J. 2023. "Social Cost of Carbon Estimates Have Increased over Time." *Nature Climate Change* 13 (6): 532–36.
- Watson, Charles C., and Mark E. Johnson. 2004. "Hurricane Loss Estimation Models: Opportunities for Improving the State of the Art." *Bulletin of the American Meteorological Society* 85 (11): 1713–26.
- Zhai, Alice R., and Jonathan H. Jiang. 2014. "Dependence of US Hurricane Economic Loss on Maximum Wind Speed and Storm Size." *Environmental Research Letters: ERL [Web Site]* 9 (6): 064019.

REVIEWERS' COMMENTS

Reviewer #3 (Remarks to the Author):

I was asked to step in for Reviewer #1 to judge whether the authors had appropriately answered the requests and concerns made by Reviewer #1. While I find the comments made by R#1 for the most part valid, the authors have done a magnificent job in responding to them. They have replied in great detail and have put a lot of effort into changing their manuscript accordingly. Overall, the paper has greatly benefited from the changes made. I have only one minor additional point of short-coming worth adding to the discussion: There are also other ways a tropical cyclone could harm the economy besides wind speed – storm surge and tropical cyclone-related rainfall also negatively affect the economy. Hence, the estimates of the authors should only be seen as a lower-bound estimate of the “true” damage by tropical cyclones.

Response to Reviews

Reviewer #3:

I was asked to step in for Reviewer #1 to judge whether the authors had appropriately answered the requests and concerns made by Reviewer #1. While I find the comments made by R#1 for the most part valid, the authors have done a magnificent job in responding to them. They have replied in great detail and have put a lot of effort into changing their manuscript accordingly. Overall, the paper has greatly benefited from the changes made. I have only one minor additional point of short-coming worth adding to the discussion: There are also other ways a tropical cyclone could harm the economy besides wind speed – **storm surge and tropical cyclone-related rainfall also negatively affect the economy. Hence, the estimates of the authors should only be seen as a lower-bound estimate of the “true” damage by tropical cyclones.**

Response:

We thank the reviewer for stepping in and keeping the review process alive. We would fully agree that it is important to add to the discussion that there are other important drivers of damages than wind fields. Indeed, we have been working over the last two years towards a better representation of storm surge and rainfall and are now working on including these drivers in our damage estimates. We have added the following sentences to the discussion:

Furthermore, our damage estimates are based solely on the wind fields of the storms, and we neglect TC-related rainfall (Bakkensen et al., 2018) and storm surge (Neumann et al., 2015) as drivers of TC damage. Since the storm-surge risk increases with rising sea levels, and also TC rainfall is projected to intensify under global warming (Gori et al., 2022; Knutson et al., 2020), our damage projections should be considered as a lower bound of the overall expected damage from TCs.

References

- Bakkensen, L. A., Park, D.-S. R., & Sarkar, R. S. R. (2018). Climate costs of tropical cyclone losses also depend on rain. *Environmental Research Letters: ERL [Web Site]*, 13(7), 074034.
- Gori, A., Lin, N., Xi, D., & Emanuel, K. (2022). Tropical cyclone climatology change greatly exacerbates US extreme rainfall–surge hazard. *Nature Climate Change*, 12(2), 171–178.
- Knutson, T., Camargo, S. J., Chan, J. C. L., Emanuel, K., Ho, C.-H., Kossin, J., Mohapatra, M., Satoh, M., Sugi, M., Walsh, K., & Wu, L. (2020). Tropical Cyclones and Climate Change Assessment: Part II: Projected Response to Anthropogenic Warming. *Bulletin of the American Meteorological Society*, 101(3), E303–E322.
- Neumann, J. E., Emanuel, K., Ravela, S., Ludwig, L., Kirshen, P., Bosma, K., & Martinich, J. (2015). Joint effects of storm surge and sea-level rise on US Coasts: new economic estimates of impacts, adaptation, and benefits of mitigation policy. *Climatic Change*, 129(1-2), 337–349.